# HopCast: Calibration of Autoregressive Dynamics Models

**Muhammad Bilal Shahid**  *belal@iastate.edu*
*Department of Mechanical Engineering*
*Iowa State University*
**Cody Fleming**  *flemingc@iastate.edu*
*Department of Mechanical Engineering*
*Iowa State University*

**Reviewed on OpenReview:** *https: // openreview. net/ forum? id= wsO6nxvGof*

## Abstract

Deep learning models are often trained to approximate dynamical systems that can be modeled using differential equations. Many of these models are optimized to predict one-step ahead; such approaches produce calibrated one-step predictions if the predictive model can quantify uncertainty, such as Deep Ensembles. At inference time, multi-step predictions are generated via autoregression, which needs a sound uncertainty propagation method to produce calibrated multi-step predictions. This work introduces an alternative Predictor-Corrector approach named HOPCAST that uses Modern Hopfield Networks (MHN) to learn the errors of a deterministic Predictor that approximates the dynamical system. The Corrector predicts a set of errors for the Predictor's output based on a context state at any timestep during autoregression. The set of errors creates sharper and well-calibrated prediction intervals with higher predictive accuracy compared to baselines without uncertainty propagation. The calibration and prediction performances are evaluated across a set of dynamical systems. This work is also the first to benchmark existing uncertainty propagation methods based on calibration errors. We also evaluate HOPCAST as a substitute for Deep Ensembles within a model-based reinforcement learning planner, demonstrating improved performance across multiple control tasks.

## 1 Introduction

Approximating dynamical systems that can be modeled using differential equations has many applications in systems biology (Wang et al., 2021; Brauer & Kribs, 2016), control (Moerland et al., 2023; Lu et al., 2021), economics (Tu, 2012), and cyber-physical systems (Bilal Shahid & Fleming, 2025; Shahid et al., 2024; Robison et al., 2024). Many deep learning approaches to approximate dynamical systems optimize the model to predict one-step ahead during training (Khansari-Zadeh & Billard, 2011; Coates et al., 2008; Ko et al., 2007). The multi-step predictions are generated via autoregression at inference time (Lu et al., 2021). In doing so, the model error accumulates over time, resulting in multi-step predictions diverging from the ground truth (Venkatraman et al., 2015; Janner et al., 2021). Hence, it is imperative to develop methods to generate accurate and calibrated multi-step predictions.

State-of-the-art methods for generating accurate and calibrated predictions include Probabilistic Ensembles (Deep Ensembles) (Lakshminarayanan et al., 2017). To approximate the dynamics typically associated with differential equations, these models are optimized to predict one-step ahead during training and then autoregressively generate multi-step predictions at inference time. To generate calibrated multi-step predictions, Probabilistic Ensembles need sound uncertainty propagation methods such as Trajectory Sampling (Chua et al., 2018a).

In contrast, we introduce a Predictor-Corrector mechanism to produce accurate and calibrated multi-step predictions for dynamical systems without the need for uncertainty propagation. The Predictor is optimized

to predict a one-step ahead point estimate of the next state of the system during training and produce multi-step predictions via autoregression at inference time. Our Corrector is based on the idea that similar states of a system lead to similar errors. This idea was leveraged by Auer et al. (2023) to build calibrated prediction intervals for one-step prediction utilizing Modern Hopfield Networks (MHN). However, the same state may not lead to the same error for multi-step predictions for dynamical systems. In such situations, the error also depends on how the system arrives at that state, i.e., where the system started and when it reached the state. To that end, we introduce the concept of *context state*, assuming that similar context states lead to similar errors. The Corrector, utilizing MHN, learns similarity between context states in terms of errors. At any timestep during autoregression, the Predictor predicts, and the Corrector generates prediction intervals for the prediction. To control the width of prediction intervals for calibration, we introduce the concept of Attention Span. We name our approach HopCast.

Our contributions are:

- the introduction of the Predictor-Corrector mechanism, where the Predictor autoregressively generates the next state for a dynamical system, and the Corrector predicts a set of errors for the full forecasting horizon. The expected error corrects the prediction, while the set of errors generates the calibrated prediction intervals.

- a method – HopCast – that has no assumptions about the Predictor except that it predicts a point estimate of the next state of the system. It is therefore a general framework that can provide multi-step calibrated predictions irrespective of the form or capability of the underlying Predictor.

- lower calibration and prediction error across several benchmarks, without the use of complex uncertainty propagation techniques. Unlike uncertainty propagation methods, HopCast creates sharper prediction intervals based on context state instead of accumulating uncertainty over time during autoregression. We also test HopCast as a replacement for Probabilistic Ensembles in a model-based reinforcement learning planner, achieving strong performance on standard control tasks.

## 2 Related Work

Since we propose a mechanism to quantify the predictive uncertainty and generate calibrated prediction intervals, the related works include those from the uncertainty quantification (UQ) and model calibration. In UQ, there are two lines of work: Bayesian and Ensemble methods.

**Bayesian Methods**. In Bayesian models, a distribution over the parameters of the underlying model is learned, called posterior distribution. At inference time, multiple samples of parameters can be taken from the posterior distribution using an accurate sampling method, such as Hamiltonian Monte Carlo (HMC) (Betancourt, 2017). These samples serve as multiple predictive models and give us the desired diversity in the predictive variable of interest. A notable method from this domain is Bayes by Backprop (Blundell et al., 2015), which learns an approximate posterior distribution over the weights of a feedforward model via Variational Inference (VI) by optimizing the evidence lower bound (ELBO) (Kingma et al., 2015).

**Ensemble Methods**. In classical *ensemble* methods, an ensemble of models is trained with variations in the data for each model in the ensemble to train robust predictors (Davison & Hinkley, 1997). Deep Ensembles, however, are constructed using modern deep learning models with random initializations in the parameter space. These over-parametrized models, when initialized randomly, converge to different local minima, giving us the desired diversity in the predictive variable (Fort et al., 2019) These models were first presented by Lakshminarayanan et al. (2017) in the context of predictive uncertainty estimation. A notable extension of that is Neural Ensemble Search (Zaidi et al., 2021), which uses different architectures for models in an ensemble and improves upon deep ensembles. The Deep Ensemble is a scalable way of quantifying uncertainty compared to Bayesian methods (Wilson & Izmailov, 2020). Hence, these are used as baseline models with different uncertainty propagation methods in the current work.

**Model Calibration**. Calibration has been studied widely for classification models. In binary classification, Platt Scaling (Platt et al., 1999) and isotonic regression (Niculescu-Mizil & Caruana, 2005) have been used

successfully for recalibration. There are extensions of such works to multi-class classification problems (Zadrozny & Elkan, 2002). In the context of regression, Gneiting & Raftery (2007) proposed several proper scoring rules to evaluate the calibration of a probabilistic model for continuous variables. Those scoring rules have been used in the literature as loss functions, for example, continuous ranked probability score (CRPS) (Gasthaus et al., 2019). Calibration has also been discussed in the literature on probabilistic forecasting, mainly in the context of meteorology (Gneiting & Raftery, 2005), resulting in specialized calibration systems (Raftery et al., 2005). An approach called calibrated regression was proposed by Kuleshov et al. (2018), which used isotonic regression to recalibrate Bayesian models. In contrast, our work turns a deterministic Predictor into a calibrated Predictor via a Predictor-Corrector mechanism. Also, we focus on Predictors that *autoregressively* generate the next state of a dynamical system, a setting rarely addressed in the model calibration literature to the best of our knowledge.

## 3 Problem Description

Consider a dynamical system described by a multivariate ordinary differential equation (ODE).

$$\dot{\mathbf{x}}(t) = \frac{d\mathbf{x}(t)}{dt} = \mathbf{f}(\mathbf{x}(t)) \tag{1}$$

where $\mathbf{x}(t) \in \mathbb{R}^D$ denotes the state of a $D$-dimensional system at time $t$ and $\dot{\mathbf{x}}(t) \in \mathbb{R}^D$ is its first order time derivative. The $\mathbf{f}(\mathbf{x}(t))$ specifies the vector-valued time derivative function. The state of a dynamical system at time $t$ can be obtained by integrating the ODE as:

$$\mathbf{x}(t) = \mathbf{x}_0 + \int_0^t \mathbf{f}(\mathbf{x}(\tau))d\tau \tag{2}$$

The ODE is integrated forward in time starting from the initial condition $\mathbf{x}(0) = \mathbf{x}_0$. We assume that the vector field $\mathbf{f}$ is unknown, but it can be learned based on observed data. The goal of a predictive model is to predict $\mathbf{x}(t)$ as accurately as possible, for as long as possible, that is, to minimize the distance between the prediction $\hat{\mathbf{x}}(t)$, and ground-truth trajectory, $\mathbf{x}(t)$. Additionally, in this work, for $D$ predictions of each variable in $\mathbf{x}(t)$, we wish to create a set of models that correct such predictions, $M_i(t)$ for $i \in \{1, \ldots, D\}$.

The following exposition describes the data generation process and nomenclature for Predictors, Correctors, trajectory types, errors, and so forth. The method is general and can be used for any $D$-dimensional problem as in equation (1). For ease of exposition and without loss of generality, we select a two-dimensional ODE to describe the inner workings and preliminaries. The Lotka-Volterra (LV) system is used here and in section 4 for illustrative purposes; subsequent sections show experimental results for a variety of systems. LV dynamics can be written as:

$$\frac{dx}{dt} = \alpha x - \beta xy \tag{3}$$

$$\frac{dy}{dt} = \delta xy - \gamma y \tag{4}$$

The state of this system can be defined as: $\tilde{\mathbf{s}} = (x, y)$. We assume a set of ground-truth $N$ trajectories of $T$ timesteps $I_N = \{\tau_n\}_{n=1}^N$ where $\tau_n = \{(\mathbf{s}_t^n, \mathbf{s}_{t+1}^n)_t\}_{t=1}^T$ and $\mathbf{s}_t^n = (x_t^n, y_t^n)$. This dataset can be generated by solving the equations with a numerical ODE solver such as `solve_ivp` from `scipy` with random initial conditions. To simulate measurement noise, we perturb each state post-integration with additive Gaussian noise, i.e., $\mathbf{s} = \tilde{\mathbf{s}} + \epsilon$, where $\epsilon \sim \mathcal{N}(0, \sigma * \Sigma)$ and $\Sigma = \text{diag}(\sigma_x, \sigma_y)$. The $\sigma$ is the scaling factor. We name $I_N$ an *Integrated* dataset. We assume that there is a Predictor $B$ (a learned representation of vector-valued function $\mathbf{f}$), trained on $I_N$, that predicts $\bar{\mathbf{s}}_{t+1}$ [1] given $\bar{\mathbf{s}}_t$, autoregressively generating a predicted set of $N$ trajectories starting from the same initial conditions. Precisely, $A_N = \{\bar{\tau}_n\}_{n=1}^N$ where $\bar{\tau}_n = \{(\bar{\mathbf{s}}_t^n, \bar{\mathbf{s}}_{t+1}^n)_t\}_{t=1}^T$ and $\bar{\mathbf{s}}_t^n = (\bar{x}_t^n, \bar{y}_t^n)$. We call $A_N$ an *Autoregressive* dataset. The error $(\mathbf{e}_t^n)$ is the difference between $t^{th}$

---

[1]The overhead bar over a variable shows that it belongs to *Autoregressive* dataset.

timestep of $n^{th}$ trajectory from *Integrated* and *Autoregressive*, that is, $\mathbf{e}_t^n = (e_{x_t^n}, e_{y_t^n})$ where $\mathbf{e}_t^n = \mathbf{s}_t^n - \bar{\mathbf{s}}_t^n$. Therefore, the set of errors of $N$ trajectories is $F_N = \{\{\mathbf{e}_t^n\}_{t=1}^T\}_{n=1}^N$.

For the LV example, we assume two correction models, $M_x$ and $M_y$, corresponding to each output of the Predictor $B$. $M_x$ and $M_y$ will predict a set of errors $E_{\bar{x}} = \{\varepsilon_{\bar{x}_{t+1}}^i\}_{i=1}^s$ and $E_{\bar{y}} = \{\varepsilon_{\bar{y}_{t+1}}^i\}_{i=1}^s$ associated with $t^{th}$ timestep of a trajectory at inference time, respectively. The $z\%$ prediction interval, i.e., $\mathrm{PI}_{\bar{x}_{t+1}}^z$ and $\mathrm{PI}_{\bar{y}_{t+1}}^z$ for $\bar{x}_{t+1}$ and $\bar{y}_{t+1}$ at $t^{th}$ timestep can be generated as follows:

$$\mathrm{PI}_{\bar{x}_{t+1}}^z = \bar{x}_{t+1} + [Q_\alpha(E_{\bar{x}}), Q_{1-\alpha}(E_{\bar{x}})] \tag{5}$$

$$\mathrm{PI}_{\bar{y}_{t+1}}^z = \bar{y}_{t+1} + [Q_\alpha(E_{\bar{y}}), Q_{1-\alpha}(E_{\bar{y}})] \tag{6}$$

where $\alpha = \frac{1-z}{2}$. For example, a 90% prediction interval corresponds to $\alpha = 0.05$. The $Q_\alpha$ denotes the $\alpha$-quantile. We generate prediction intervals for the outputs of Predictor $B$ at $t^{th}$ timestep during autoregression solely based on the set of errors predicted by correction models ($M_x$ and $M_y$), thereby avoiding uncertainty propagation.

## 4 HopCast

The proposed methodology, HopCast, consists of a Predictor-Corrector mechanism shown in Fig. 1. At any timestep ($t$) during autoregression, the Predictor $B$ produces a point forecast of the next state of the system ($\bar{x}_{t+1}, \bar{y}_{t+1}$) given previous state ($\bar{x}_t, \bar{y}_t$). The Corrector retrieves context-dependent errors ($E_{\bar{x}}$ and $E_{\bar{y}}$) to correct the forecast and quantify uncertainty. In particular, the Corrector consists of correction models ($M_x$ and $M_y$), one for each output of the Predictor $B$. Each correction model ($M$) consists of an Encoder ($m$) and an MHN (Ramsauer et al., 2020; Auer et al., 2023). The Encoder ($m$) is a fully connected feedforward model. Architectural details are included in Appendix E.1. We frame correction as a **pattern retrieval task** (Ramsauer et al., 2020). Precisely, $M_x$ and $M_y$ take in the context state ($\bar{x}_1, \bar{y}_1, \bar{x}_t, \bar{y}_t, t$) as a pattern and outputs $E_{\bar{x}}$ and $E_{\bar{y}}$, respectively. $E_{\bar{x}}$ and $E_{\bar{y}}$ contain errors associated with similar context states from the past. The context state is the input of Predictor $B$ ($\bar{x}_t, \bar{y}_t$) augmented with the initial condition ($\bar{x}_1, \bar{y}_1$) and time step ($t$). The context state captures both local and trajectory-level structure, enabling the Corrector to retrieve errors specific to the current dynamical regime. The $E_{\bar{x}}$ & $E_{\bar{y}}$ are used to derive prediction intervals.

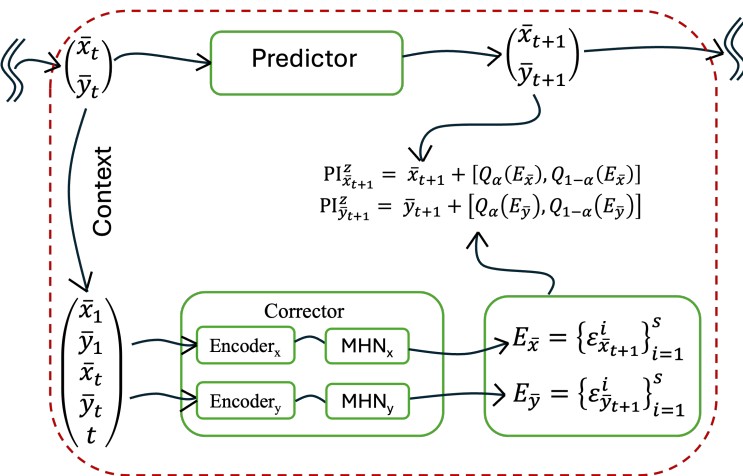

Figure 1: Predictor-Corrector mechanism

The remainder of this section consists of three subsections. Section 4.1 discusses the shuffling of $I_N$, $A_N$, & $F_N$ datasets to train correction models for pattern retrieval tasks. Section 4.2 then discusses the training of the correction model ($M_x$), and section 4.3 describes the inference with $M_x$ where we generate the set of errors ($E_{\bar{x}}$) given the context state.

### 4.1 Shuffled Data

The MHN in HOPCAST is trained as a *pattern–retrieval* model rather than a *sequence* model. To this end, we break every trajectory into independent context–error pairs and shuffle them, so the model learns to retrieve errors based on the similarity of context, rather than the temporal ordering of the data. We explain that below.

**Shuffled Queries** $(\mathcal{S}_Q)$. Before training the correction model $(M_x)$, the states of the system from *Autoregressive* $(A_N)$ dataset are augmented with context information, that is, initial condition and time step. Appendix E.2 emphasizes the importance of adding initial condition as a context. Specifically, the $t^{th}$ timestep of $n^{th}$ trajectory $\bar{\tau}_n$ from $A_N$, i.e., $(\bar{x}_t^n, \bar{y}_t^n)$, will be $\bar{\mathbf{c}}_t^n = (\bar{x}_1^n, \bar{y}_1^n, \bar{x}_t^n, \bar{y}_t^n, t)$ after adding context information. The tuple $(\bar{x}_1^n, \bar{y}_1^n)$ is the initial condition of trajectory $\bar{\tau}_n$. The $\bar{\mathbf{c}}_t^n$ denotes the context state associated with the $t^{th}$ timestep of $n^{th}$ trajectory. The set of all context states constructed from dataset $A_N$ is $\{\{\bar{\mathbf{c}}_t^n\}_{t=1}^T\}_{n=1}^N$. This nested set of context states is flattened and shuffled randomly to form a new set $\mathcal{S}_Q = \text{Shuffle}\left(\bigcup_{n=1}^N \{\bar{\mathbf{c}}_t^n\}_{t=1}^T\right)$. Appendix E.7 highlights the importance of shuffling the data.

**Shuffled Keys** $(\mathcal{S}_K)$. Likewise, we construct the context states $\{\{\mathbf{c}_t^n\}_{t=1}^T\}_{n=1}^N$ for *Integrated* $(I_N)$ dataset as well, where $\mathbf{c}_t^n = (x_1^n, y_1^n, x_t^n, y_t^n, t)$. Similar to $\mathcal{S}_Q$, we flatten the nested set $\{\{\mathbf{c}_t^n\}_{t=1}^T\}_{n=1}^N$ and randomly shuffle it to form $\mathcal{S}_K = \text{Shuffle}\left(\bigcup_{n=1}^N \{\mathbf{c}_t^n\}_{t=1}^T\right)$.

**Shuffled Values** $(\mathcal{S}_{V_x} \ \& \ \mathcal{S}_{V_y})$. Given errors $F_N = \{\{\mathbf{e}_t^n\}_{t=1}^T\}_{n=1}^N$ where $\mathbf{e}_t^n = (e_{x_t^n}, e_{y_t^n})$, $F_N$ can be split into two sets of errors $F_N^x = \{\{e_{x_t^n}\}_{t=1}^T\}_{n=1}^N$ and $F_N^y = \{\{e_{y_t^n}\}_{t=1}^T\}_{n=1}^N$. Recall that we train separate correction models $(M_x$ and $M_y)$ for each output of Predictor $B$. $F_N^x$ and $F_N^y$ are flattened and shuffled to form $\mathcal{S}_{V_x} = \text{Shuffle}\left(\bigcup_{n=1}^N \{e_{x_t^n}\}_{t=1}^T\right)$ and $\mathcal{S}_{V_y} = \text{Shuffle}\left(\bigcup_{n=1}^N \{e_{y_t^n}\}_{t=1}^T\right)$.

Once we have $\mathcal{S}_Q$, $\mathcal{S}_K$, $\mathcal{S}_{V_x}$, $\mathcal{S}_{V_y}$, they are split into train/test with an 80/20 split. To train $M_x$, we need dataset $\mathcal{D}_x = \{(\mathbf{Q}, \mathbf{K}, \mathbf{V}_x)_i\}$ comprising triplets $(\mathbf{Q}, \mathbf{K}, \mathbf{V}_x)$. Each triplet of $\mathbf{Q}$, $\mathbf{K}$, and $\mathbf{V}_x$ are constructed by sampling $\mathtt{S_L}$ (Sequence Length) number of elements from sets $\mathcal{S}_Q^{\text{Train}}$, $\mathcal{S}_K^{\text{Train}}$, and $\mathcal{S}_{V_x}^{\text{Train}}$, respectively. For instance, one triplet with $\mathtt{S_L} = 5$ is: $\mathbf{Q} = (\bar{\mathbf{c}}_4^3, \bar{\mathbf{c}}_5^1, \bar{\mathbf{c}}_7^5, \bar{\mathbf{c}}_2^3, \bar{\mathbf{c}}_1^7)^\top$, $\mathbf{K} = (\mathbf{c}_4^3, \mathbf{c}_5^1, \mathbf{c}_7^5, \mathbf{c}_2^3, \mathbf{c}_1^7)^\top$, and $\mathbf{V}_x = (e_{x_4^3}, e_{x_5^1}, e_{x_7^5}, e_{x_2^3}, e_{x_1^7})^\top$. The error $e_{x_4^3} = x_4^3 - \bar{x}_4^3$ denotes the error of Predictor $B$ at $4^{th}$ timestep of $3^{rd}$ trajectory. Likewise, we construct $\mathcal{D}_y = \{(\mathbf{Q}, \mathbf{K}, \mathbf{V}_y)_i\}$ using $\mathcal{S}_Q^{\text{Train}}$, $\mathcal{S}_K^{\text{Train}}$ and $\mathcal{S}_{V_y}^{\text{Train}}$. The 20% split with inference data is used for final evaluation.

### 4.2 Training

Figure 2 illustrates how the correction model $(M_x)$ retrieves context-dependent errors using an Encoder $(m_x)$ and a Modern Hopfield Network $(\text{MHN}_x)$: (i) the $m_x$ maps each context state into a $d$-dimensional embedding, and (ii) the $\text{MHN}_x$ forms associations between embedded queries and keys and retrieves the corresponding error values. Specifically, the $m_x$ takes in as input the context state, such as $\mathbf{c}_4^3$, and outputs a $d$-dimensional embedding. We choose $d = 4$ in our experiments, though other values of $d$ are expected to perform comparably [Appendix E.6]. We use the same $m_x$ to construct embedded queries $\mathbf{Q}_\psi$ and keys $\mathbf{K}_\psi$ from $\mathbf{Q}$ and $\mathbf{K}$, respectively. These appear as row/column labels in the *Training* block of Fig. 2. The $\text{MHN}_x$ uses an attention mechanism (Vaswani, 2017) to construct an association matrix $(\mathbf{A}_{\mathtt{S_L} \times \mathtt{S_L}})$ based on the similarity of elements in $\mathbf{Q}_\psi$ with elements in $\mathbf{K}_\psi$. Mathematically,

$$\mathbf{A}_{\mathtt{S_L} \times \mathtt{S_L}} = \texttt{softmax}\left(\frac{\mathbf{Q}_\psi \mathbf{K}_\psi^\top}{\sqrt{d}}\right) \tag{7}$$

$$\hat{\mathbf{V}}_x = \mathbf{A}_{\mathtt{S_L} \times \mathtt{S_L}} . \mathbf{V}_x \tag{8}$$

$$\mathcal{L} = \frac{1}{\mathtt{S_L}} \|\mathbf{V}_x - \hat{\mathbf{V}}_x\|_2^2 \tag{9}$$

The association matrix ($\mathbf{A}_{S_L \times S_L}$) is shown in the *Training* block of Fig. 2 with $S_L = 5$. We mask the association from *Autoregressive* context state to its *Integrated* context state. This masking is reflected in the zeroed diagonal entries of the association matrix. The $(i,j)^{th}$ entry of $\mathbf{A}_{S_L \times S_L}$ shows the similarity of $i^{th}$ element of $\mathbf{Q}_\psi$ with $j^{th}$ element of $\mathbf{K}_\psi$. We use $\mathtt{softmax}$ over $\mathbf{A}_{S_L \times S_L}[i,:]$ to get association weights for $i^{th}$ element of $\mathbf{Q}_\psi$. Therefore, each row of $\mathbf{A}_{S_L \times S_L}$ sums up to one. The loss function $\mathcal{L}$ is used to learn the parameters of the $m_x$ via backpropagation, where the $\hat{\mathbf{V}}_x$ denotes the predicted errors.

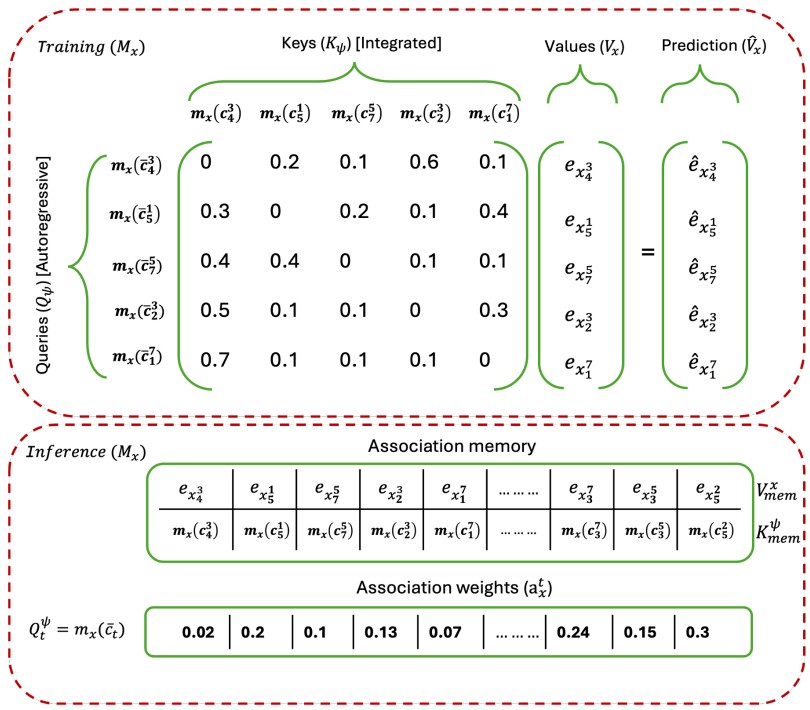

Figure 2: Training and Inference with Correction Model (Encoder$_x$ + MHN$_x$).

## 4.3 Inference

At inference time, the correction model ($M_x$) retrieves past errors based on the similarity between the encoded query and the encoded keys stored in the MHN$_x$ Association memory. To build the Association memory, a set of $K$ keys ($\mathbf{K}_{mem}$) with the corresponding values ($\mathbf{V}_{mem}^x$) is sampled from $\mathcal{S}_K^{\mathrm{Train}}$ and $\mathcal{S}_{V_x}^{\mathrm{Train}}$, respectively. For instance, a randomly sampled set of keys and values would be $\mathbf{K}_{mem} = (\mathbf{c}_4^3, \mathbf{c}_5^1, \mathbf{c}_7^5, \cdots, \mathbf{c}_5^2)$ and $\mathbf{V}_{mem}^x = (e_{x_4^3}, e_{x_5^1}, e_{x_7^5}, \cdots, e_{x_5^2})$, respectively. The trained Encoder $m_x$ is used to encode the $\mathbf{K}_{mem}$ as $\mathbf{K}_{mem}^\psi$. The $\mathbf{K}_{mem}^\psi$ and $V_{mem}^x$ constitute the Association memory of MHN$_x$. It is shown in the *Inference* block in Fig. 2. We loaded 2000 keys in memory for evaluation [Appendix E.5]. Once the memory is set up, the correction model ($M_x$) takes in the context state ($\bar{x}_1, \bar{y}_1, \bar{x}_t, \bar{y}_t, t$) as a query $\mathbf{Q}_t$ at $t^{th}$ time step during autoregression and encodes it as $\mathbf{Q}_t^\psi$. The association weights ($\mathbf{a}_x^t$) at $t^{th}$ timestep are,

$$\mathbf{a}_x^t = \mathtt{softmax}\left(\frac{\mathbf{Q}_t^\psi \mathbf{K}_{mem}^\psi}{\sqrt{d}}\right) \qquad (10)$$

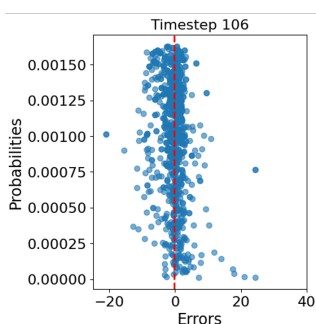

Figure 3: An example set of errors ($E_x$) sampled from Association Memory at Timestep 106 during autoregression for the x-output of the Predictor for the Lorenz system, where the red dotted line denotes the ground truth error.

where $\mathbf{a}_x^t = (p_1, \cdots, p_K)$, $\sum_{i=1}^K p_i = 1$, and $K$ denotes the number of keys in memory. The weights $\mathbf{a}_x^t$ show how strongly the query $\mathbf{Q}_t^\psi$ matches each entry in $\mathbf{K}_{mem}^\psi$. A set of $s$ categories is sampled as: $\{b_i\}_{i=1}^s \sim \text{Categorical}(\mathbf{a}_x^t)$. We select $s = 1000$ in our experiments [Appendix E.3]. A set of errors $(E_{\bar{x}})$ is selected from $\mathbf{V}_{mem}^x$ based on sampled categories $\{b_i\}_{i=1}^s$, i.e., $E_{\bar{x}} = \{V_{mem}^x[b_i]\}_{i=1}^s$. A sampled set of errors $E_{\bar{x}}$ is shown in Fig. 3 for the x-output of the Predictor $B$ for the Lorenz system. An alternative approach to sampling from $\mathbf{a}_x^t$ is to retrieve top-$k$ probabilities. We provide an ablation study on this in Appendix E.4.

We have discussed the training of $M_x$ and how to utilize it to obtain a set of errors $(E_{\bar{x}})$ for the x-output of the Predictor $B$ at the inference time. To obtain $E_{\bar{y}}$ by training $M_y$, we repeat the same procedure stated in section 4.2 and section 4.3 with dataset $\mathcal{D}_y$, and this can be done for an arbitrary number of dimensions depending on the system dynamics.

## 5 Experiments

### 5.1 Evaluation Metrics

We propose three metrics to evaluate the quality of HopCast against the benchmarks.

**Calibration Error (Kuleshov et al., 2018)** (CE). A calibrated model has observed fractions ($\hat{p}$) of its predicted random variable match with the expected fractions ($p$). The difference between these two is calibration error. To evaluate CE, $w(= 9)$ equally spaced prediction intervals are chosen from 10% to 90%. For each prediction interval, we count the number of times the observed variable falls between intervals. Mathematically, $\text{CE} = \sum_{i=1}^w (\hat{p}_i - p_i)^2$.

**Prediction Interval Width (Auer et al., 2023)** (PI-Width). A calibrated model should also be sharp. Sometimes, it is trivial to reduce the CE by always predicting the expected value of a random variable (Kuleshov et al., 2018). A good predictor should be calibrated and sharp. Hence, we propose to use PI-Width as a measure of sharpness. Mathematically,

$$\text{PI-Width} = \frac{1}{w}\frac{1}{v}\sum_{i=1}^w \sum_{j=1}^v |U_j^i - L_j^i|, \tag{11}$$

where $w = 9$ is for equally spaced PI, and $v$ denotes the number of validation data points; i.e., $v = |\mathcal{S}_Q^{\text{Inference}}|$.

**Mean Squared Error** (MSE). MSE is used to evaluate the predictive accuracy of the proposed approach against the baselines.

### 5.2 Datasets

We have discussed one dynamical system, i.e., LV, in section 3. Other dynamical systems include Lorenz, FitzHugh-Nagumo (FHN), Lorenz95, and the Glycolytic Oscillator. To generate datasets, we randomly sample $N$ initial conditions from within a specified range for the state variables of each system. The `solve_ivp` method from `scipy` is used to integrate the dynamics with the adaptive-step RK45 solver. To produce uniformly sampled trajectories, system states are extracted at fixed time intervals $\Delta t$ as mentioned in Table 4. For Lorenz and LV, the dynamics of both system states and their derivatives are modeled, whereas the dynamics of states are modeled for the rest of the systems. The mathematical forms of each dynamical system, ranges of initial conditions, and parameter values are given in Appendix B.

### 5.3 Baselines

The baseline methods include three uncertainty propagation approaches, i.e., Expectation, Moment Matching, and Trajectory Sampling. These approaches are used to propagate uncertainty with Probabilistic Ensembles (Chua et al., 2018a). The details of Probabilistic Ensembles' training and uncertainty propagation methods are presented in Appendix A.

# 6 Results

**Overall Comparison** As shown in Table 1, HOPCAST outperforms the baseline approaches in terms of CE and MSE in the vast majority of cases. It achieves the lowest average CE of 0.046 among all, with the second-best approach being the Trajectory Sampling with CE of 0.075. In terms of MSE, HOPCAST outperforms in 11 out of 15 cases, showing the effectiveness of our Predictor-Corrector mechanism. The Expectation and Moment Matching do not perform well in terms of CE and MSE. These two approaches generate conservative intervals with average PI-Widths of 7.56 and 3.63, respectively, resulting in overconfident models. The overconfidence is evident in the calibration curves of Expectation and Moment Matching in Fig. 6. The average CE of these two approaches is 0.21 and 0.68, respectively, showing high miscalibration compared to Trajectory Sampling and HOPCAST. PI-Width is only useful when discussed in conjunction with CE as it is trivial to reduce the PI-Width while being overconfident and miscalibrated.

Table 1: Prediction Interval Widths (**PI-Widths**) and Calibration Error (**CE**) metrics of HOPCAST and baseline approaches with different scaling factor ($\sigma$) across various dynamical systems. The results are averaged across 3 runs of the same experiment. All of the CEs within 5% of the best one in a row are highlighted. The PI-Width is only used as a tie-breaker if there are different models with CEs within 5% of the best. In that case, the best-performing model based on CE is denoted with an asterisk and PI-Width is highlighted. For MSE, we simply highlight anything within 5% of the best one in a row.

| Model | | HopCast | | | Probabilistic Ensemble (PE) | | | | | | | | |
| --- | --- | --- | --- | --- | --- | --- | --- | --- | --- | --- | --- | --- | --- |
| | | | | | Expectation | | | Moment Matching | | | Trajectory Sampling | | |
| Metrics | | MSE | PI-Width | CE | MSE | PI-Width | CE | MSE | PI-Width | CE | MSE | PI-Width | CE |
| Lotka Volterra | $\sigma=0.05$ | $6.87\pm0.32$ | $2.93\pm0.17$ | $0.022\pm0.004$ | $20.58\pm1.25$ | $4.01\pm0.40$ | $\mathbf{0.017\pm0.009}$ | $28.9\pm1.04$ | $0.65\pm0.06$ | $0.56\pm0.12$ | $19.45\pm0.36$ | $5.43\pm0.10$ | $0.22\pm0.11$ |
| | $\sigma=0.1$ | $6.10\pm0.20$ | $1.88\pm0.05$ | $\mathbf{0.0070\pm0.002}$ | $24.1\pm0.15$ | $0.83\pm0.02$ | $0.47\pm0.17$ | $23.9\pm0.04$ | $0.96\pm0.04$ | $0.41\pm0.14$ | $21.01\pm0.27$ | $5.87\pm0.59$ | $0.18\pm0.10$ |
| | $\sigma=0.3$ | $9.69\pm0.23$ | $3.19\pm0.12$ | $\mathbf{0.0051\pm0.004}$ | $25.6\pm0.09$ | $2.13\pm0.02$ | $0.21\pm0.05$ | $25.4\pm0.08$ | $2.93\pm0.02$ | $0.059\pm0.02$ | $23.3\pm0.05$ | $6.19\pm0.22$ | $0.13\pm0.07$ |
| Lorenz | $\sigma=0.05$ | $1126\pm7.97$ | $28.76\pm0.74$ | $0.011\pm0.006$ | $\mathbf{1101.9\pm28.3}$ | $30.37\pm0.80$ | $\mathbf{0.002\pm0.001}$ | $1991.3\pm4.03$ | $4.45\pm0.009$ | $1.23\pm0.07$ | $\mathbf{1080.10\pm5.59}$ | $38.79\pm0.49$ | $0.16\pm0.04$ |
| | $\sigma=0.1$ | $1248.17\pm15$ | $38.66\pm0.38$ | $0.019\pm0.03$ | $1413.1\pm38.83$ | $35.49\pm1.11$ | $\mathbf{0.011\pm0.015}$ | $2579.7\pm48.7$ | $8.78\pm0.01$ | $1.21\pm0.08$ | $1314.60\pm3.80$ | $45.28\pm0.69$ | $0.078\pm0.03$ |
| | $\sigma=0.3$ | $1681\pm46.36$ | $\mathbf{40.44\pm0.91}$ | $\mathbf{0.019\pm0.004^*}$ | $1848.5\pm15.8$ | $25.9\pm1.43$ | $0.38\pm0.10$ | $2098.5\pm18.08$ | $23.13\pm0.06$ | $0.58\pm0.30$ | $1788.6\pm10.9$ | $48.52\pm0.35$ | $\mathbf{0.018\pm0.016}$ |
| FHN | $\sigma=0.05$ | $\mathbf{0.076\pm0.002}$ | $0.20\pm0.005$ | $0.091\pm0.004$ | $0.68\pm0.13$ | $1.13\pm0.061$ | $\mathbf{0.044\pm0.034}$ | $0.95\pm0.45$ | $0.28\pm0.007$ | $0.91\pm0.42$ | $0.66\pm0.04$ | $1.25\pm0.007$ | $0.054\pm0.026$ |
| | $\sigma=0.1$ | $\mathbf{0.17\pm0.03}$ | $1.03\pm0.05$ | $0.32\pm0.05$ | $1.39\pm0.17$ | $0.76\pm0.064$ | $0.38\pm0.19$ | $2.21\pm0.13$ | $0.38\pm0.003$ | $1.42\pm0.086$ | $0.83\pm0.005$ | $1.35\pm0.003$ | $\mathbf{0.051\pm0.02}$ |
| | $\sigma=0.3$ | $\mathbf{0.57\pm0.04}$ | $1.27\pm0.03$ | $0.11\pm0.02$ | $1.53\pm0.086$ | $1.009\pm0.046$ | $0.39\pm0.13$ | $1.25\pm0.003$ | $1.52\pm0.010$ | $0.108\pm0.019$ | $1.12\pm0.008$ | $1.55\pm0.004$ | $\mathbf{0.061\pm0.011}$ |
| Lorenz95 | $\sigma=0.05$ | $10.44\pm0.08$ | $4.41\pm0.13$ | $\mathbf{0.007\pm0.003}$ | $10.11\pm0.21$ | $3.25\pm0.015$ | $0.13\pm0.017$ | $14.11\pm0.48$ | $1.84\pm0.009$ | $0.62\pm0.04$ | $\mathbf{9.30\pm0.031}$ | $4.90\pm0.023$ | $0.022\pm0.007$ |
| | $\sigma=0.1$ | $13.98\pm0.11$ | $5.78\pm0.018$ | $\mathbf{0.0028\pm0.007}$ | $13.31\pm0.20$ | $3.29\pm0.05$ | $0.27\pm0.025$ | $17.38\pm1.14$ | $3.24\pm0.03$ | $0.35\pm0.06$ | $\mathbf{11.63\pm0.11}$ | $5.44\pm0.04$ | $0.004\pm0.003$ |
| | $\sigma=0.3$ | $16.44\pm0.66$ | $6.08\pm0.06$ | $\mathbf{0.0015\pm0.001}$ | $17.46\pm0.11$ | $4.61\pm0.04$ | $0.13\pm0.029$ | $16.37\pm0.23$ | $6.01\pm0.03$ | $0.0036\pm0.003$ | $\mathbf{15.13\pm0.075}$ | $6.49\pm0.047$ | $0.009\pm0.006$ |
| Glycolytic Oscillator | $\sigma=0.05$ | $\mathbf{0.03\pm0.001}$ | $0.20\pm0.007$ | $\mathbf{0.043\pm0.018}$ | $0.10\pm0.006$ | $0.24\pm0.02$ | $0.07\pm0.04$ | $0.13\pm0.019$ | $0.056\pm0.002$ | $1.35\pm0.32$ | $0.09\pm0.004$ | $0.36\pm0.006$ | $0.06\pm0.026$ |
| | $\sigma=0.1$ | $\mathbf{0.072\pm0.003}$ | $0.25\pm0.007$ | $\mathbf{0.018\pm0.006}$ | $0.12\pm0.005$ | $0.21\pm0.01$ | $0.10\pm0.03$ | $0.15\pm0.005$ | $0.08\pm0.004$ | $0.99\pm0.18$ | $0.10\pm0.001$ | $0.41\pm0.01$ | $0.07\pm0.03$ |
| | $\sigma=0.3$ | $\mathbf{0.11\pm0.001}$ | $\mathbf{0.37\pm0.008}$ | $\mathbf{0.016\pm0.003^*}$ | $0.18\pm0.001$ | $0.23\pm0.01$ | $0.62\pm0.28$ | $0.25\pm0.002$ | $0.21\pm0.003$ | $0.54\pm0.07$ | $0.17\pm0.005$ | $0.46\pm0.003$ | $\mathbf{0.015\pm0.01}$ |
| Average | - | - | 9.03 | 0.046 | - | 7.56 | 0.21 | - | 3.63 | 0.68 | - | 11.48 | 0.075 |

**Attention Span** HOPCAST lets us control the confidence of the model with precise control over the width of PI based on a concept we introduce called *Attention Span*. The *Attention Span* can be increased/decreased by increasing/decreasing the $\mathsf{S_L}$ of $\mathbf{Q}$ and $\mathbf{K}$ in $\mathbf{A}_{\mathsf{S_L}\times\mathsf{S_L}}$. To demonstrate that, we collect a dataset $H = \{(x_i, y_i)\}_{i=1}^{6000}$ using the sine function with heteroscedastic noise (Chua et al., 2018a) shown in equation 12 and Fig. 4(a). The $x_i$'s are uniformly sampled from the following domain $[-\frac{5\pi}{2}, -\pi] \cup [\pi, \frac{5\pi}{2}]$.

$$(x, y) \rightarrow \left(x, y + \mathcal{N}\left(0, 0.2 \left|\sin\left(\frac{3}{2}x + \frac{\pi}{8}\right)\right|\right)\right) \tag{12}$$

We use the same correction model ($M_x$) consisting of Encoder ($m_x$) and MHN$_x$ with the same training and inference procedures described in section 4.2 and 4.3, respectively. The only exception is that the set of triplets $\{(\mathbf{Q}, \mathbf{K}, \mathbf{V})_i\}$ are constructed from dataset $H$. For instance, a triplet with $\mathsf{S_L} = 4$ is: $\mathbf{Q} =$

$(x_{10}, x_{13}, x_{14}, x_1)$, $\mathbf{K} = (x_{10}, x_{13}, x_{14}, x_1)$, and $\mathbf{V} = (y_{10}, y_{13}, y_{14}, y_1)$. We trained two $M_x$ models with $\mathtt{S_L}$ of 3 & 8. At the inference time, the $M_x$ is given a data point $x = 3.28$ as a query to be associated with the keys in memory based on similarity in terms of $y$. The keys from memory are sampled based on their association weights. The frequency plots of sampled keys for $\mathtt{S_L}$ of 3 & 8 are in Fig. 4(b) & (c), respectively. For $\mathtt{S_L} = 3$, the sampled keys are clustered around 3.28 in Fig. 4(b), which are the most similar keys to the query $x = 3.28$. As we increase $\mathtt{S_L}$ to 8, the $M_x$ expands its *Attention Span* by picking other keys from memory around $x = -3.2 \,\&\, 6.5$ that have nearly similar $y$ values as shown in Fig. 4(c).

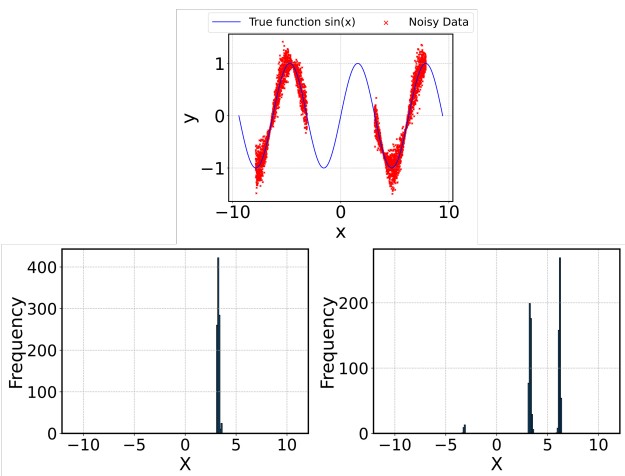

Figure 4: (a) Upper center: Sine function with heteroscedastic noise (b) Lower left: Attention Span for $x = 3.28$ as a query with $\mathtt{S_L} = 3$ (c) Lower right: Attention Span for $x = 3.28$ as a query with $\mathtt{S_L} = 8$

**Calibration Performance** The reason HopCast performs well in terms of CE compared to baselines is owing to the $\mathtt{S_L}$ that can be increased/decreased to increase/decrease the diversity of sampled errors $E_{\bar{x}}$ from MHN memory. As we increase the $\mathtt{S_L}$, the $M_x$ starts to expand its *Attention Span* by associating other keys from memory ($\mathbf{K}_{men}^{\psi}$) to the query ($\mathbf{Q}_t^{\psi}$) based on their similarity in terms of errors, effectively controlling the width of prediction intervals. We train separate correction models (e.g., $M_x$ and $M_y$) for each output of the Predictor and tune their $\mathtt{S_L}$ separately. The $\mathtt{S_L}$ hyperparameter and number of models in a Probabilistic Ensemble for each setting are shown in Appendix C. For the baseline approaches, we tune the number of models in an ensemble. In general, we observed that the Probabilistic Ensembles tend to increase the uncertainty (or widen their prediction intervals) as we increase the number of models, and the uncertainty plateaus at some point. Therefore, if the uncertainty plateaus and the model is overconfident, adding more models to the ensemble will not make it a calibrated model. This was observed in the calibration curves of Expectation and Moment Matching in Fig. 6. Trajectory Sampling, in contrast, provided calibrated uncertainty with few models in the ensemble (i.e., 3 or 4) in most cases. It is owing to its nature of taking predicted uncertainty into account during propagation.

**Predictive Performance** HopCast outperforms in most cases in terms of MSE because we proposed a *Corrector* that predicts a set of errors, and prediction intervals are a side effect of the approach. Also, our model generates a smaller PI-width in most cases because we generate prediction intervals based on the context state $(x_0, y_0, x_t, y_t, t)$ at any timestep during autoregression. On the other hand, uncertainty propagation approaches generate intervals based on uncertainty accumulated due to propagation up to a certain time step ($t = t'$) starting from $t = 1$. A manifestation of that is shown in Fig. 5 (b) & (d). For Fig. 5(a), the output of Predictor $B$ is shown in the dark, and the Mean indicates the prediction after adding the expected error ($\sum_{i=1}^{s} \mathbf{a}_x^t[b_i] E_x[i]$) from the Corrector. For Fig. 5(b), (c), & (d), the Mean indicates the expected prediction from all models in a Probabilistic ensemble. Fig. 5(a) shows that HopCast reduces its PI-Width around time step 270 due to smaller error, resulting in sharper intervals. On the other hand, Expectation and Trajectory Sampling have wider PI due to accumulated uncertainty up to that time step, even though the Mean is close to the ground truth.

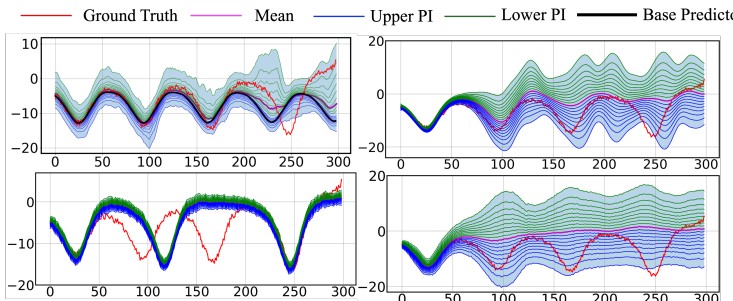

Figure 5: (a). Upper left: HOPCAST (b). Upper right: Expectation (c). Lower left: Moment Matching (d). Lower right: Trajectory Sampling. A comparison of prediction intervals generated by HOPCAST and three uncertainty propagation approaches. The x-axis denotes the time steps, and the y-axis shows the $x$ output of the Lorenz system. The Upper PI and Lower PI show the $m = 9$ equally spaced prediction intervals from 10% to 90%.

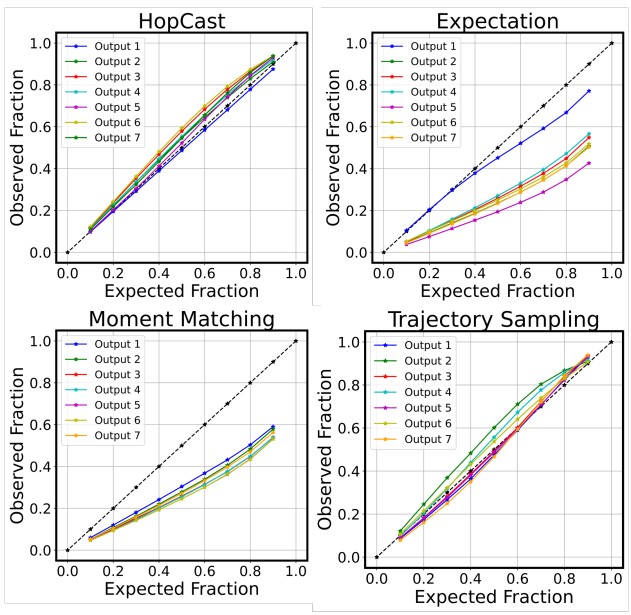

Figure 6: The calibration curves of HOPCAST and three uncertainty propagation methods for Glycolytic Oscillator at $\sigma = 0.3$. The dark dotted line corresponds to the perfect calibration. The calibration curves are shown separately for seven states of the system.

**Trajectory Sampling vs. Counterparts** On average, Trajectory Sampling performs better than its counterpart baselines–Expectation and Moment Matching–in terms of CE, owing to how it propagates uncertainty. It takes into account the predicted uncertainty of the model while propagating uncertainty from time step $t$ to $t+1$. It samples the state for the next step from the predicted Multivariate Normal distribution rather than propagating just the predicted mean like the Expectation. Moment Matching also samples the next state from the predicted Multivariate Normal distribution, just like Trajectory Sampling. However, it recasts prediction from all models in an ensemble at every timestep ($t$) to a Multivariate Normal distribution and then resamples next states before propagation. This results in overly conservative PI, as shown in Fig. 5(c), and evidenced by the PI-Width of 3.63 in Table 1.

## 7 Hyperparameter Tuning

$S_L$ (Sequence Length) is a hyperparameter that must be tuned for calibrated uncertainty. We propose Algorithm 1 to tune it based on the intuition that uncertainty typically increases with larger $S_L$ (Section 6; Fig. 7). A small value (e.g., 10) produces overconfident estimates, while a large value (e.g., 1000) leads to underconfidence. Intermediate values such as $S_L = 500$ yield near-perfect calibration, whereas $S_L = 400$ and $S_L = 600$ show overconfidence and underconfidence, respectively. We iteratively refine $S_L$ with the calibration curve and typically converge to an optimal setting within a few trials. Appendix E.9 provides additional experiments.

The number of models in an ensemble is a hyperparameter. As discussed in the section 6, the ensembles tend to build up the uncertainty until it plateaus [Appendix E.8]. We keep adding models to the ensemble until it provides calibrated uncertainty. In some cases, the uncertainty saturates while the model is overconfident. In those cases, the number of optimal models is very high, e.g., 8 or 7. Appendix F contains additional details about hyperparameters.

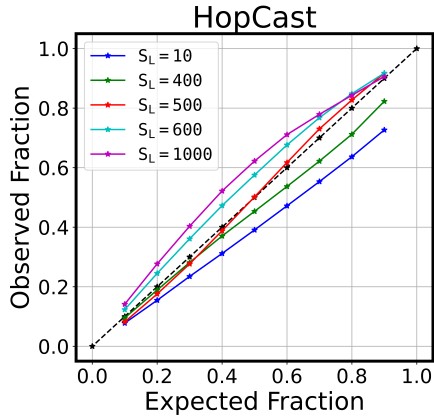

Figure 7: Calibration curves of the $y$-output of Lorenz for different $S_L$.

---

**Algorithm 1** $S_L$ Tuning for Calibration

1: **Initialize** OptimalSL ← True
2: ▷ *Pick a large* $S_L$ *& a small* $S_L$
3: ▷ *Pick an* $S_L$ *between large and small* $S_L$
4: **while** OptimalSL **do**
5:   ▷ *Three possibilities on Calibration Curve*
6:   **if** *Calibrated Uncertainty* **then**
7:     OptimalSL ← False
8:   **else if** *Underconfidence* **then**
9:     Decrease $S_L$
10:   **else if** *Overconfidence* **then**
11:     Increase $S_L$
12:   **end if**
13: **end while**

---

## 8 Model-Based Reinforcement Learning

We compare the performance of HOPCAST and Probabilistic Ensembles as the uncertainty-aware dynamics models within a model-based reinforcement learning (RL) algorithm. PETS (Probabilistic Ensembles with Trajectory Sampling) is a widely used model-based RL algorithm that relies on Probabilistic Ensembles to model the transition dynamics and propagate uncertainty for planning via Model Predictive Control (MPC) (Chua et al., 2018b). In PETS, the planning quality is heavily influenced by the accuracy and calibration of the dynamics model, model bias compounds rapidly during long-horizon rollouts, and miscalibrated dynamics models lead the planner to choose overly optimistic or overly conservative action sequences.

Our earlier experiments in Table 1 demonstrate that HOPCAST produces more accurate and better-calibrated multi-step predictions than Probabilistic Ensembles, which utilize different uncertainty propagation methods, i.e., Expectation (E) A.1, Moment Matching (MM) A.3, and Trajectory Sampling (TS) A.2. In Table 2, we show the performance of HOPCAST on four standard control tasks when it is used as a drop-in replacement for the Probabilistic Ensembles in the PETS algorithm. The results for baselines, i.e., Probabilistic Ensembles & Deterministic Ensembles, with three uncertainty propagation methods, i.e., Expectation, Moment Matching, & Trajectory Sampling, are taken from Chua et al. (2018b) [2]. The HOPCAST yields comparable performance

---

[2]Code available at: https://github.com/kchua/handful-of-trials

to Probabilistic Ensembles on Cartpole and 7-DOF Reacher, while it outperforms on 7-DOF Pusher and Half-Cheetah tasks. The Probabilistic Ensembles, which utilize uncertainty propagation methods, outperform on Cartpole and 7-DOF Reacher, showing their competitiveness against HopCast. However, the Deterministic Ensembles exhibit competitive performance in a relatively low-dimensional setting, such as Cartpole, but show performance degradation on high-dimensional tasks. These experiments demonstrate the efficacy of our proposed methodology HopCast within a model-based RL algorithm on various control tasks.

Table 2: The performance of different tasks in terms of rewards with different dynamics models, i.e., Probabilistic Ensembles (PE) & Deterministic Ensembles (DE), and various uncertainty propagation methods, i.e., Expectation (E), Moment Matching (MM), & Trajectory Sampling (TS). The HopCast shows the results with our proposed methodology as a drop-in replacement of PE within the PETS planning algorithm. The best results are shown in bold.

| Task | Model | | | | | | |
|------|-------|-----|-------|-------|------|-------|-------|
| | HopCast | PE-E | PE-MM | PE-TS | DE-E | DE-MM | DE-TS |
| Cartpole | $181 \pm 2.3$ | 180 | 181 | **183** | 179 | 177 | 181 |
| 7-DOF Pusher | $\mathbf{-45 \pm 4.3}$ | $-48$ | $-46$ | $-46$ | $-95$ | $-97$ | $-93$ |
| 7-DOF Reacher | $-44 \pm 3.4$ | $-44$ | $-45$ | $\mathbf{-43}$ | $-93$ | $-94$ | $-96$ |
| Half-cheetah | $\mathbf{7170 \pm 40}$ | 5700 | 200 | 7100 | 3800 | 190 | 3950 |

## 9 Conclusions

We proposed a Predictor-Corrector mechanism for autoregressive dynamics models to correct the Predictors' predictions and generate calibrated prediction intervals for it at any timestep during autoregression. HopCast performs competitively well against the alternative approach (i.e., uncertainty propagation), giving accurate and calibrated autoregressive dynamics models. Out of three uncertainty propagation approaches, i.e., Trajectory Sampling, Moment Matching, and Expectation, Trajectory Sampling performs competitively well against ours. This is due to its nature of taking predicted uncertainty into account during propagation. HopCast offers a lower calibration and prediction error with sharper prediction intervals. The sharper prediction intervals result directly from modeling context-specific errors, and the lower prediction error is the consequence of modeling errors in the form of a Corrector. In addition, we introduce a concept called *Attention Span* that gives precise control over the width of prediction intervals with a hyperparameter we introduce called Sequence Length ($S_L$). We also deploy HopCast within a model-based reinforcement learning planner as an alternative to Probabilistic Ensembles, demonstrating its competitive performance on diverse control tasks.

**Limitations and Future Directions** One of the limitations of our work is due to the use of explicit timestep IDs (i.e., $t = 1,2,3$, etc.) in the context state ($c_t^n$). At inference time, the Corrector will only generate calibrated prediction intervals for a trajectory length equal to or less than what it was trained on. Moreover, the Corrector was trained on trajectories sampled at a fixed time interval $\Delta t$ and does not generalize to other sampling intervals. A modified version of the Corrector could be made discretization-invariant. We hand-tuned the $S_L$ hyperparameter, training multiple models to get a lower calibration error. An alternative approach to obtaining calibrated uncertainty with HopCast without tuning the $S_L$ hyperparameter is an interesting area for future work.

## 10 Acknowledgements

This work was partially supported by the National Aeronautics and Space Administration, USA, under grant 80NSSC24CA037.

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

# A Baseline Methods

This section will discuss how we propagate uncertainty with Probabilistic Ensembles using three uncertainty propagation methods, i.e., Expectation. Moment Matching and Trajectory Sampling. These will be discussed with reference to the Lotka-Volterra (LV) system, whose state at time step $(t + 1)$ is $\mathbf{s}_{t+1} = (x_{t+1}, y_{t+1})$. We will see how to generate $z\%$ prediction intervals associated with $\mathbf{s}_{t+1}$ ($\mathrm{PI}^z_{\mathbf{s}_{t+1}}$) at any timestep during autoregression with these approaches.

**Probabilistic Ensembles**   To construct Probabilistic Ensembles, we train a population of fully connected feedforward models, i.e., $\Omega = \{f_i\}_i$, where $f_i$ represents the $i^{th}$ model within the population. Each model is trained with Gaussian negative log-likelihood (NLL) loss as demonstrated in Lakshminarayanan et al. (2017). Each model predicts a Multivariate Normal (MVN) over the next state $\mathbf{s}_{t+1}$ given the previous state of the system $\mathbf{s}_t$. Formally, $\mathbf{s}_{t+1} \sim \mathrm{MVN}(\boldsymbol{\mu}_{t+1}, \boldsymbol{\Sigma}_{t+1})$. For the LV system, $\boldsymbol{\mu}_{t+1} = (\mu^x_{t+1}, \mu^y_{t+1})$ and $\boldsymbol{\Sigma}_{t+1} = \mathrm{diag}(\sigma^x_{t+1}, \sigma^y_{t+1})$.

**Uncertainty Propagation**   At inference time, a sample of $M$ models $\{f_m\}_{m=1}^M$ can be taken from the population ($\Omega$) to form a Probabilistic Ensemble. All three propagation approaches use a particle-based approach, where the core idea is to generate *multiple trajectories* via autoregression starting from the *same initial condition* (Chua et al., 2018a). Let $\mathbf{s}_t^{p,m}$ denote the state of the system associated with $p^{th}$ particle from $m^{th}$ model at $t^{th}$ time step. Each model $f_m$ receives $P$ copies of the same initial condition $\{\mathbf{s}_{t=0}^{p,m}\}_{p=1}^P$, where $P$ denotes the number of particles assigned to model $f_m$. The propagation of uncertainty starts with $P * M$ copies of the same initial conditions across $\{f_m\}_{m=1}^M$ models and results in $P * M$ number of trajectories. The diversity in these trajectories will come from two sources, i.e., $M$ different models and their probabilistic predictions. These diverse trajectories will be used to derive prediction intervals. In contrast, HOPCAST used a set of errors (e.g., $E_{\bar{x}}$ & $E_{\bar{y}}$) to derive prediction intervals.

The rest of the section discusses the algorithmic details of three propagation approaches and the construction of prediction intervals using the outputs of three algorithms.

## A.1  Expectation

The Expectation method always uses $P = 1$ particle for each model $f_m$. This is because the predicted uncertainty ($\boldsymbol{\Sigma}_{t+1}^{p,m}$) is ignored, and the predicted mean ($\boldsymbol{\mu}_{t+1}^{p,m}$) is propagated to the next time step. Therefore, assigning more than one particle to a model $f_m$ will not generate diverse trajectories from that model. Even though this approach ignores predicted uncertainty ($\boldsymbol{\Sigma}_{t+1}^{p,m}$) during propagation, it will be used to generate prediction intervals as explained in section A.3. The overall procedure of this approach is in Algorithm 2.

---

**Algorithm 2** Expectation

1: **Input:** Initial states $\{\mathbf{s}_{t=0}^{1,m}\}_{m=1}^M$, models $f_1, \ldots, f_M$, timesteps $T$
2: **for** $t = 0$ **to** $T - 1$ **do**
3:    **for** $m = 1$ **to** $M$ **do**
4:       Predict $\boldsymbol{\mu}_{t+1}^{1,m}, \boldsymbol{\Sigma}_{t+1}^{1,m} \leftarrow f_m(\mathbf{s}_t^{1,m})$
5:       $\mathbf{s}_{t+1}^{1,m} = \boldsymbol{\mu}_{t+1}^{1,m}$  ▷ *Propagate mean*
6:    **end for**
7:    $\{s_{t+1}^{1,m}\}_{m=1}^M$  ▷ *inputs to the next timestep*
8: **end for**
9: **return** $\{\mathbf{s}_{1:T}^{1,m}\}_{m=1}^M$  ▷ *the $1 * M$ number of trajectories of length $T$*

---

## A.2  Trajectory Sampling

This method uses $P$ particles for each model $f_m$ since it considers predicted uncertainty ($\boldsymbol{\Sigma}_{t+1}^{p,m}$) while propagating uncertainty to the next step. The overall procedure of this algorithm is summarized in Algorithm 3.

---

**Algorithm 3** Trajectory Sampling

---

1: **Input:** Initial states $\{\{\mathbf{s}_{t=0}^{p,m}\}_{p=1}^{P}\}_{m=1}^{M}$, models $f_1, \ldots, f_M$, timesteps $T$, particles $P$
2: **for** $t = 0$ **to** $T - 1$ **do**
3:     **for** $m = 1$ **to** $M$ **do**
4:         **for** $p = 1$ **to** $P$ **do**
5:             Predict $\boldsymbol{\mu}_{t+1}^{p,m}, \boldsymbol{\Sigma}_{t+1}^{p,m} \leftarrow f_m(\mathbf{s}_t^{p,m})$
6:             Sample $\mathbf{s}_{t+1}^{p,m} \sim \text{MVN}(\boldsymbol{\mu}_{t+1}^{p,m}, \boldsymbol{\Sigma}_{t+1}^{p,m})$  ▷ *considers predicted uncertainty*
7:         **end for**
8:     **end for**
9:     $\{\{s_{t+1}^{p,m}\}_{p=1}^{P}\}_{m=1}^{M}$  ▷ *inputs to the next timestep*
10: **end for**
11: **return** $\{\{\mathbf{s}_{1:T}^{p,m}\}_{p=1}^{P}\}_{m=1}^{M}$  ▷ *the $P * M$ number of trajectories of length $T$*

---

## A.3   Moment Matching

This method uses $P$ particles for each model $m$ since it considers predicted uncertainty ($\boldsymbol{\Sigma}_t^{p,m}$) while propagating uncertainty to the next step. This is different from Trajectory Sampling in that it fits a Gaussian distribution to $P * M$ predictions $\{\{\mathbf{s}_{t+1}^{p,m}\}_{p=1}^{P}\}_{m=1}^{M}$ at $t^{th}$ timestep, that is, $\text{MVN}(\boldsymbol{\mu}_{t+1}, \boldsymbol{\Sigma}_{t+1})$. We assume independence between all $P * M$ predictions. A set of $P * M$ samples $\{\{\mathbf{s}_{t+1}^{p,m}\}_{p=1}^{P}\}_{m=1}^{M}$ is taken from that distribution which becomes input at the next time step. The overall procedure is summarized in Algorithm 4.

---

**Algorithm 4** Moment Matching

---

1: **Input:** Initial states $\{\{\mathbf{s}_0^{p,m}\}_{p=1}^{P}\}_{m=1}^{M}$, models $f_1, \ldots, f_M$, timesteps $T$, particles $P$
2: **for** $t = 0$ **to** $T - 1$ **do**
3:     **for** $m = 1$ **to** $M$ **do**
4:         **for** $p = 1$ **to** $P$ **do**
5:             Predict $\boldsymbol{\mu}_{t+1}^{p,m}, \boldsymbol{\Sigma}_{t+1}^{p,m} \leftarrow f_m(\mathbf{s}_t^{p,m})$
6:             Sample $\mathbf{s}_{t+1}^{p,m} \sim \text{MVN}(\boldsymbol{\mu}_{t+1}^{p,m}, \boldsymbol{\Sigma}_{t+1}^{p,m})$
7:         **end for**
8:     **end for**
9:     $\{\{\mathbf{s}_{t+1}^{p,m}\}_{p=1}^{P}\}_{m=1}^{M}$             ▷ fit a Gaussian distribution
10:     $\boldsymbol{\mu}_{t+1} \leftarrow \frac{1}{P \cdot M} \sum_{m=1}^{M} \sum_{p=1}^{P} \mathbf{s}_{t+1}^{p,m}$  ▷ *Mean*
11:     $\boldsymbol{\Sigma}_{t+1} \leftarrow \frac{1}{P \cdot M} \sum_{m=1}^{M} \sum_{p=1}^{P} (\mathbf{s}_{t+1}^{p,m} - \boldsymbol{\mu}_{t+1})(\mathbf{s}_{t+1}^{p,m} - \boldsymbol{\mu}_{t+1})^T$  ▷ *Variance*
12:     $\{\{\mathbf{s}_{t+1}^{p,m}\}_{p=1}^{P}\}_{m=1}^{M} \sim \text{MVN}(\boldsymbol{\mu}_{t+1}, \boldsymbol{\Sigma}_{t+1})$  ▷ sample inputs for the next timestep
13: **end for**
14: **return** $\{\{\mathbf{s}_{1:T}^{p,m}\}_{p=1}^{P}\}_{m=1}^{M}$  ▷ *the $P * M$ number of trajectories of length $T$*

---

**Prediction intervals using uncertainty propagation**    All three approaches output $P * M$ number of trajectories of length $T$ $\{\{\mathbf{s}_{1:T}^{p,m}\}_{p=1}^{P}\}_{m=1}^{M}$. At any time step $t$ during propagation, the $(\text{PI}_{\mathbf{s}_{t+1}}^z)$ are generated using $\{\{\mathbf{s}_t^{p,m}\}_{p=1}^{P}\}_{m=1}^{M}$. Specifically, we construct a $\text{MVN}_{net}$ whose parameters are: $\boldsymbol{\mu}_{net} = \frac{1}{P * M} \sum_{m=1}^{M} \sum_{p=1}^{P} \mathbf{s}_t^{p,m}$ and $\boldsymbol{\Sigma}_{\text{net}} = \frac{1}{P * M} \sum_{m=1}^{M} \sum_{p=1}^{P} \left[ \boldsymbol{\Sigma}_t^{p,m} + (\mathbf{s}_t^{p,m} - \boldsymbol{\mu}_{\text{net}})(\mathbf{s}_t^{p,m} - \boldsymbol{\mu}_{\text{net}})^T \right]$. For LV system, the $\boldsymbol{\mu}_{net} = (\mu_{net}^x, \mu_{net}^y)$ and $\Sigma_{net} = \text{diag}(\sigma_{net}^x, \sigma_{net}^y)$. The prediction intervals for $x$ and $y$ are derived as follows:

$$\text{PI}_{\mathbf{x}_{t+1}}^z = \mu_{net}^x \pm \Phi^{-1}(1 - \alpha)\sigma_{net}^x \tag{13}$$

$$\text{PI}_{\mathbf{y}_{t+1}}^z = \mu_{net}^y \pm \Phi^{-1}(1 - \alpha)\sigma_{net}^y \tag{14}$$

where $\alpha = \frac{1-z}{2}$ for $z\%$ prediction intervals. The $\Phi^{-1}$ denotes the inverse cumulative distribution function of the Standard Normal distribution. In contrast, equations 5 & 6 were used to generate prediction intervals with HOPCAST.

## B Dynamical Systems

There are five dynamical systems studied in this paper. One of those is Lotka-Volterra (LV) equations, which are already discussed in section 3 along with its closed form, parameters, and ranges of initial conditions. Here, we repeat the description of the LV system and then describe the rest of the systems. We add zero-mean additive Gaussian noise to each differential equation post-integration as discussed in section 3, with the variance equal to the standard deviation of the variable modelled by the differential equation scaled by a factor $\sigma$.

### B.1 Lotka-Volterra (Wangersky, 1978)

$$\frac{dx}{dt} = \alpha x - \beta xy \tag{15}$$

$$\frac{dy}{dt} = \delta xy - \gamma y \tag{16}$$

**Parameters**: $\alpha = 1.1$; $\beta = 0.4$; $\gamma = 0.4$; $\delta = 0.1$
**Initial Condition Ranges**: $x \in [5, 20]$; $y \in [5, 10]$

### B.2 Lorenz (Brunton et al., 2016)

$$\frac{dx}{dt} = \sigma(y - x) \tag{17}$$

$$\frac{dy}{dt} = x(\rho - z) - y \tag{18}$$

$$\frac{dz}{dt} = xy - \beta z \tag{19}$$

**Parameters**: $\sigma = 10$; $\rho = 28$; $\beta = \frac{8}{3}$
**Initial Condition Ranges**: $x \in [-20, 20]$; $y \in [-20, 20]$; $z \in [0, 50]$

### B.3 FitzHugh-Nagumo (FHN) (Izhikevich & FitzHugh, 2006)

$$\frac{dv}{dt} = v - \frac{v^3}{3} - w + I \tag{20}$$

$$\frac{dw}{dt} = \epsilon(v + a - bw) \tag{21}$$

**Parameters**: $a = 0.7$; $b = 0.8$; $\epsilon = 0.08$; $I = 0.5$
**Initial Condition Ranges**: $v \in [-1.5, 1.5]$; $w \in [-1.5, 1.5]$

### B.4 Lorenz95 (Lorenz, 1996)

$$\frac{dX_1}{dt} = (X_2 - X_5)X_4 - X_1 + F \tag{22}$$

$$\frac{dX_2}{dt} = (X_3 - X_1)X_5 - X_2 + F \tag{23}$$

$$\frac{dX_3}{dt} = (X_4 - X_2)X_1 - X_3 + F \tag{24}$$

$$\frac{dX_4}{dt} = (X_5 - X_3)X_2 - X_4 + F \tag{25}$$

$$\frac{dX_5}{dt} = (X_1 - X_4)X_3 - X_5 + F \tag{26}$$

**Parameters**: $F = 8$
**Initial Condition Ranges**: $X_i \in [-10.5, 10.5]$ where $i \in \{1, 2, 3, 4, 5\}$

### B.5 Glycolytic Oscillator (Daniels & Nemenman, 2015)

$$\frac{dS_1}{dt} = J_0 - \frac{k_1 S_1 S_6}{1 + (S_6/K_1)^q} \tag{27}$$

$$\frac{dS_2}{dt} = 2\frac{k_1 S_1 S_6}{1 + (S_6/K_1)^q} - k_2 S_2(N - S_5) - k_6 S_2 S_5 \tag{28}$$

$$\frac{dS_3}{dt} = k_2 S_2(N - S_5) - k_3 S_3(A - S_6) \tag{29}$$

$$\frac{dS_4}{dt} = k_3 S_3(A - S_6) - k_4 S_4 S_5 - \kappa(S_4 - S_7) \tag{30}$$

$$\frac{dS_5}{dt} = k_2 S_2(N - S_5) - k_4 S_4 S_5 - k_6 S_2 S_5 \tag{31}$$

$$\frac{dS_6}{dt} = -2\frac{k_1 S_1 S_6}{1 + (S_6/K_1)^q} + 2k_3 S_3(A - S_6) - k_5 S_6 \tag{32}$$

$$\frac{dS_7}{dt} = \psi\kappa(S_4 - S_7) - kS_7 \tag{33}$$

**Parameters**: $J_0 = 2.5; k_1 = 100; k_2 = 6; k_3 = 16; k_4 = 100; k_5 = 1.28; k_6 = 12; k = 1.8; \kappa = 13; q = 4; K_1 = 0.52; \psi = 0.1; N = 1; A = 4$
**Initial Condition Ranges**: $S_1 \in [0.15, 1.60]; S_2 \in [0.19, 2.16]; S_3 \in [0.04, 0.20]; S_4 \in [0.10, 0.35]; S_5 \in [0.08, 0.30]; S_6 \in [0.14, 2.67]; S_7 \in [0.05, 0.10]$

## C Hyperparameters

In Table 3, we provide the Sequence Length ($S_L$) for each output of the Predictor and a number of models in Probabilistic Ensembles for each experimental setting in Table 1.

## D Data Generation

Table 4: Details about data generation

| Model | $\Delta t$ | Timesteps | Trajectories |
|---|---|---|---|
| Lotka Volterra | 0.1 | 300 | 500 |
| Lorenz | 0.01 | 300 | 1000 |
| FHN | 0.5 | 400 | 350 |
| Lorenz95 | 0.01 | 300 | 666 |
| Glycolytic | 0.01 | 400 | 750 |

## E Ablation Studies

### E.1 Encoder Architecture

As discussed in section 4, each correction model utilizes an Encoder with MHN. The architecture of the Encoder (e.g., $m_x$) is a fully connected feedforward network with one layer and 100 neurons for all experiments. Here, we show the impact of adding more layers of fully connected neurons to the Encoder on the proposed metrics. Table 5 shows that adding more layers doesn't significantly impact the MSE and CE. However, the MSE and CE increase considerably when the Encoder is removed from the correction models.

Table 3: Sequence lengths ($S_L$) for all outputs of Predictor and number of models in Probabilistic ensemble across various dynamical systems and noise levels

| Model | | HopCast | Probabilistic Ensemble | | |
|---|---|---|---|---|---|
| | | | Expectation | Moment Matching | Trajectory Sampling |
| Hyperparameter | | Sequence lengths | Models | Models | Models |
| Lotka Volterra | $\sigma = 0.05$ | 30,30,50,60 | 4 | 5 | 3 |
| | $\sigma = 0.1$ | 70,30,70,40 | 5 | 5 | 3 |
| | $\sigma = 0.3$ | 70,20,60,70 | 6 | 5 | 4 |
| Lorenz | $\sigma = 0.05$ | 30,35,35,35,50,50,100 | 6 | 8 | 3 |
| | $\sigma = 0.1$ | 500,500,500,300,300,300 | 7 | 7 | 4 |
| | $\sigma = 0.3$ | 25,25,25,500,500,500 | 8 | 7 | 3 |
| FHN | $\sigma = 0.05$ | 35,55 | 5 | 5 | 3 |
| | $\sigma = 0.1$ | 15,15 | 4 | 5 | 3 |
| | $\sigma = 0.3$ | 15,15 | 5 | 4 | 3 |
| Lorenz95 | $\sigma = 0.05$ | 2200 for all | 6 | 7 | 3 |
| | $\sigma = 0.1$ | 2000 for all | 6 | 5 | 4 |
| | $\sigma = 0.3$ | 2000 for all | 7 | 6 | 4 |
| Glycolytic Oscillator | $\sigma = 0.05$ | 35,50,35,100,35,50,30 | 8 | 6 | 3 |
| | $\sigma = 0.1$ | 50,50,35,35,800,35,100 | 6 | 7 | 3 |
| | $\sigma = 0.3$ | 50 for all | 7 | 8 | 3 |

Table 5: Effect of different Encoding network architectures on the proposed metrics. The results without the Encoder are also included. The $FC(100)_1$ means the fully connected feedforward model with one layer of 100 neurons. The results are reported over three runs of the same experiment at random seeds.

| Dynamical System | Lorenz | | | Lotka Volterra | | |
|---|---|---|---|---|---|---|
| Noise Level | $\sigma = 0.3$ | | | $\sigma = 0.3$ | | |
| Metrics | MSE | PI-Width | CE | MSE | PI-Width | CE |
| Without Encoder | $1907.07 \pm 9.33$ | $38.69 \pm 0.42$ | $0.06 \pm 0.005$ | $16.12 \pm 0.36$ | $3.37 \pm 0.23$ | $0.025 \pm 0.012$ |
| $FC(100)_1$ | $1681 \pm 46.36$ | $40.44 \pm 0.91$ | $0.019 \pm 0.004$ | $9.69 \pm 0.23$ | $3.19 \pm 0.12$ | $0.0051 \pm 0.004$ |
| $FC(100)_2$ | $1717.362 \pm 30.3$ | $41.84 \pm 0.68$ | $0.014 \pm 0.002$ | $9.43 \pm 0.23$ | $3.08 \pm 0.09$ | $0.006 \pm 0.002$ |
| $FC(100)_3$ | $1710.18 \pm 46.19$ | $42.45 \pm 0.55$ | $0.018 \pm 0.002$ | $9.37 \pm 0.007$ | $2.98 \pm 0.07$ | $0.0035 \pm 0.001$ |

## E.2 Initial condition and context state ($c_t^n$)

As discussed in section 1, our context state ($c_t^n$) comprises the initial condition, the current state of the system, and the timestep ID. Here, we show the results on two systems without the initial condition a part of the context state in Table 6. The MSE increases significantly for both cases, while the CE exhibits minor changes. This shows the significance of making the initial condition a part of the context state.

Table 6: Comparison of proposed metrics with and without initial condition included in the context state. The results are averaged over three runs.

| Dynamical System | Glycolytic Oscillator | | | Lotka Volterra | | |
|---|---|---|---|---|---|---|
| Noise Level | $\sigma = 0.1$ | | | $\sigma = 0.1$ | | |
| Metrics | MSE | PI-Width | CE | MSE | PI-Width | CE |
| With IC | $0.072 \pm 0.003$ | $0.25 \pm 0.007$ | $0.018 \pm 0.006$ | $6.10 \pm 0.20$ | $1.88 \pm 0.05$ | $0.0070 \pm 0.002$ |
| Without IC | $0.176 \pm 0.003$ | $0.416 \pm 0.003$ | $0.027 \pm 0.008$ | $43.90 \pm 1.09$ | $5.17 \pm 0.14$ | $0.011 \pm 0.002$ |

### E.3  Number of Retrieved Samples ($s$)

To generate calibrated prediction intervals, a set of $s$ samples is drawn from $\mathbf{a}_x^t$ as described in section 4.3. Here, we analyze the sensitivity of proposed metrics with varying values of $s = \{100, 500, 800, 1000\}$ for two settings from Table 1. The results everywhere else in the paper are reported with $s = 1000$. In Table 7, it can be seen that the results are largely unchanged as the number of samples varies from 1000 to 500. However, at $s = 100$, we see a drop in performance in terms of MSE and CE, suggesting that too few samples limit the diversity of error samples, negatively impacting the performance.

Table 7: Comparison of proposed metrics with varying number of retrieved residual samples ($s$) from MHN memory. The results are averaged over three runs.

| Dynamical System | Glycolytic Oscillator | | | Lotka Volterra | | |
|---|---|---|---|---|---|---|
| Noise Level | $\sigma = 0.1$ | | | $\sigma = 0.1$ | | |
| Metrics | MSE | PI-Width | CE | MSE | PI-Width | CE |
| 1000 | $0.072 \pm 0.003$ | $0.25 \pm 0.007$ | $0.018 \pm 0.006$ | $6.10 \pm 0.20$ | $1.88 \pm 0.05$ | $0.0070 \pm 0.002$ |
| 800 | $0.070 \pm 0.005$ | $0.24 \pm 0.009$ | $0.018 \pm 0.008$ | $6.10 \pm 0.2$ | $1.89 \pm 0.03$ | $0.007 \pm 0.003$ |
| 500 | $0.071 \pm 0.005$ | $0.26 \pm 0.003$ | $0.015 \pm 0.008$ | $6.13 \pm 0.20$ | $1.87 \pm 0.04$ | $0.006 \pm 0.005$ |
| 100 | $0.096 \pm 0.005$ | $0.18 \pm 0.006$ | $0.025 \pm 0.008$ | $7.9 \pm 0.17$ | $0.69 \pm 0.04$ | $0.019 \pm 0.004$ |

### E.4  Top-$k$ retrieval

As discussed in section 4.3, the $s$ samples are drawn from $\mathbf{a}_x^t$ to generate calibrated prediction intervals. Here, we analyze an alternative approach to sampling, i.e., using top-$k$ probabilities from $\mathbf{a}_x^t$. Table 8 shows the results on proposed metrics with varying values of $k = \{100, 500, 800, 1000\}$. The results remain largely unchanged for $k \geq 500$. At $s = 100$, the CE shows an increase for both settings, indicating that too few samples lead to decreased diversity in sampled errors.

### E.5  Size of Association Memory

In section 4.3, we discussed the inference with the correction model ($M_x$). A set of $K$ keys ($\mathbf{K}_{mem}$) with the corresponding values ($\mathbf{V}_{mem}^x$) is randomly sampled from $\mathcal{S}_K^{\text{Train}}$ and $\mathcal{S}_{V_x}^{\text{Train}}$, respectively and loaded into the Association memory. Here, the impact of the size of Association memory on the proposed metrics will be discussed. Table 9 shows the variation in proposed metrics as the number of keys in memory varies from 10 to 2000. For all systems, the MSE and CE improve significantly as the keys increase from 10 to 50. The MSE and CE that showed a considerable change as the number of keys increases are shown in bold. As the

Table 8: Comparison of proposed metrics based on top-$k$ probabilities from $\mathbf{a}_t^x$ instead of sampling. The results are reported for $k = \{100, 500, 800, 1000\}$. The results are averaged over three runs.

| Dynamical System | Glycolytic Oscillator | | | Lotka Volterra | | |
|---|---|---|---|---|---|---|
| Noise Level | $\sigma = 0.1$ | | | $\sigma = 0.1$ | | |
| Metrics | MSE | PI-Width | CE | MSE | PI-Width | CE |
| 1000 | $0.67 \pm 0.002$ | $0.24 \pm 0.007$ | $0.011 \pm 0.005$ | $7.31 \pm 0.99$ | $2.38 \pm 0.33$ | $0.016 \pm 0.003$ |
| 800 | $0.067 \pm 0.002$ | $0.24 \pm 0.01$ | $0.011 \pm 0.005$ | $7.52 \pm 1.18$ | $2.41 \pm 0.38$ | $0.014 \pm 0.004$ |
| 500 | $0.069 \pm 0.003$ | $0.24 \pm 0.009$ | $0.017 \pm 0.005$ | $8.39 \pm 2.50$ | $2.53 \pm 0.64$ | $0.017 \pm 0.008$ |
| 100 | $0.07 \pm 0.004$ | $0.20 \pm 0.005$ | $0.06 \pm 0.03$ | $6.36 \pm 0.22$ | $1.24 \pm 0.11$ | $0.25 \pm 0.036$ |

keys increase beyond 50, not all systems' MSE and CE showed a considerable increase. The MSE and CE show insignificant changes as the number of keys increases beyond 100 until 2000. Thus, we fix the number of keys to 2000 for our final evaluation of results.

Table 9: The results on proposed metrics with varying numbers of keys in memory. The MSE and CE that showed a considerable improvement with the increase in the number of keys in memory are shown in bold. The results are averaged over three runs.

| Keys in Memory | Metric | Lorenz | Glycolytic Oscillator | Lotka Volterra | FHN | Lorenz95 |
|---|---|---|---|---|---|---|
| | | $\sigma = 0.1$ | $\sigma = 0.3$ | $\sigma = 0.05$ | $\sigma = 0.05$ | $\sigma = 0.3$ |
| 10 | MSE | $\mathbf{1765.48 \pm 170.73}$ | $\mathbf{0.15 \pm 0.009}$ | $\mathbf{26.86 \pm 4.80}$ | $\mathbf{0.15 \pm 0.03}$ | $\mathbf{24.81 \pm 3.11}$ |
| | PI-Width | $34.59 \pm 5.69$ | $0.37 \pm 0.006$ | $3.61 \pm 1.06$ | $0.24 \pm 0.03$ | $6.59 \pm 0.69$ |
| | CE | $\mathbf{0.11 \pm 0.03}$ | $\mathbf{0.080 \pm 0.009}$ | $\mathbf{0.10 \pm 0.04}$ | $\mathbf{0.24 \pm 0.19}$ | $\mathbf{0.039 \pm 0.05}$ |
| 50 | MSE | $\mathbf{1419.29 \pm 103.78}$ | $0.13 \pm 0.001$ | $\mathbf{9.66 \pm 1.51}$ | $0.072 \pm 0.001$ | $18.10 \pm 1.75$ |
| | PI-Width | $37.64 \pm 0.55$ | $0.36 \pm 0.008$ | $2.90 \pm 0.03$ | $0.21 \pm 0.012$ | $5.98 \pm 0.11$ |
| | CE | $\mathbf{0.025 \pm 0.009}$ | $0.012 \pm 0.002$ | $\mathbf{0.038 \pm 0.003}$ | $0.12 \pm 0.02$ | $\mathbf{0.016 \pm 0.003}$ |
| 100 | MSE | $1285.90 \pm 30.9$ | $0.12 \pm 0.002$ | $7.76 \pm 0.67$ | $0.076 \pm 0.001$ | $18.57 \pm 0.51$ |
| | PI-Width | $36.83 \pm 0.39$ | $0.36 \pm 0.015$ | $2.99 \pm 0.06$ | $0.21 \pm 0.01$ | $6.09 \pm 0.15$ |
| | CE | $0.015 \pm 0.007$ | $0.018 \pm 0.007$ | $0.026 \pm 0.006$ | $\mathbf{0.111 \pm 0.003}$ | $\mathbf{0.006 \pm 0.005}$ |
| 500 | MSE | $1249 \pm 11.03$ | $0.12 \pm 0.001$ | $6.89 \pm 0.30$ | $0.076 \pm 0.002$ | $17.06 \pm 0.86$ |
| | PI-Width | $38.19 \pm 1.39$ | $0.36 \pm 0.001$ | $2.85 \pm 0.08$ | $0.21 \pm 0.006$ | $6.02 \pm 0.10$ |
| | CE | $0.020 \pm 0.004$ | $0.013 \pm 0.003$ | $0.022 \pm 0.003$ | $0.093 \pm 0.03$ | $0.0021 \pm 0.003$ |
| 1000 | MSE | $1241.72 \pm 7.55$ | $0.12 \pm 0.003$ | $7.08 \pm 0.01$ | $0.077 \pm 0.001$ | $16.91 \pm 0.74$ |
| | PI-Width | $38.37 \pm 1.91$ | $0.37 \pm 0.005$ | $2.91 \pm 0.093$ | $0.21 \pm 0.001$ | $6.13 \pm 0.09$ |
| | CE | $0.018 \pm 0.001$ | $0.017 \pm 0.004$ | $0.021 \pm 0.004$ | $0.092 \pm 0.01$ | $0.0020 \pm 0.002$ |
| 2000 | MSE | $1248.17 \pm 15$ | $0.11 \pm 0.001$ | $6.87 \pm 0.32$ | $0.076 \pm 0.002$ | $16.44 \pm 0.66$ |
| | PI-Width | $38.66 \pm 0.38$ | $0.37 \pm 0.008$ | $2.93 \pm 0.17$ | $0.20 \pm 0.005$ | $6.08 \pm 0.06$ |
| | CE | $0.019 \pm 0.03$ | $0.016 \pm 0.003$ | $0.022 \pm 0.004$ | $0.091 \pm 0.004$ | $0.0015 \pm 0.001$ |

### E.6 Embedding dimension

As discussed in section 4.2, the Encoder outputs a $d$-dimensional embedding given context state $(c_t^n)$, where $d = 4$. Here, we want to show that the performance of correction models is insensitive to the embedding dimension provided that the $S_L$ is retuned accordingly. In Table 10, the results on metrics and $S_L$ for LV at $\sigma = 0.1$ in the first row are copied from Tables 1 & 3, respectively. Table 10 shows that the CE shows a considerable increase from 0.0070 to 0.065 as we change the $d$ to 5 from 4, while the MSE shows minor changes. The last row shows the results at $d = 5$ when the $S_L$ is retuned, where the CE (i.e., 0.0098) is close to what we had earlier at $d = 4$ (i.e., 0.0070).

Table 10: The impact of change in embedding dimension on the proposed metrics. The last row shows the results when $S_L$ is retuned for the new embedding dimension ($d = 5$). The results are reported over three runs.

| Embedding Dimension ($d$) | $S_L$ of all outputs | Lotka Volterra | | |
| --- | --- | --- | --- | --- |
| | | $\sigma = 0.1$ | | |
| | | MSE | PI-Width | CE |
| $d = 4$ | [70,30,70,40] | $6.10 \pm 0.20$ | $1.88 \pm 0.05$ | $0.0070 \pm 0.002$ |
| $d = 5$ | [70,30,70,40] | $6.75 \pm 0.065$ | $1.94 \pm 0.02$ | $0.065 \pm 0.005$ |
| $d = 5$ | [5,5,70,5] | $6.98 \pm 0.115$ | $1.69 \pm 0.01$ | $0.0098 \pm 0.003$ |

### E.7 Impact of shuffled data

As discussed in section 4.1, the dataset is shuffled before forming queries, keys, and values for training. Here, we show that this shuffling brings two major benefits. One, the MSE and CE show significant improvement. Second, CE and MSE show changes with respect to the number of keys in memory that are consistent across different systems.

Table 11 shows the results on the proposed metrics for FHN and LV with and without shuffling the data. The MSE and CE increase considerably for both systems when the data is not shuffled, irrespective of the number of keys in memory. For LV, the MSE starts to decrease and CE starts to increase as we reduce the number of keys in memory from 2000 to 10. For FHN, in contrast, MSE and CE begin to grow as the number of keys goes down to 10 from 2000. On the other hand, the MSE and CE remain largely unchanged for both systems as we reduce the number of keys from 2000 to 500 when the data is shuffled. For keys less than 500, the MSE and CE show considerable change for both systems.

When the data is shuffled, the changes in MSE and CE as we reduce the number of keys show a generalizable pattern for both systems. Hence, the guidelines regarding the number of keys in the memory at the inference time are generalizable.

### E.8 Number of models in Probabilistic Ensembles

As discussed in section 7, the Probabilistic Ensembles tend to saturate uncertainty after a certain number of models in the ensemble. Here, we demonstrate that for all propagation approaches via calibration curves. As shown in Fig. 8(a) for Expectation, the uncertainty goes up from overconfidence to underconfidence as we increase the number of models in the Probabilistic Ensemble. The last three increments in the number of models (13 to 15) don't impact uncertainty much. For Moment Matching (Fig. 8(b)), the uncertainty increases with the number of models and saturates later while the model is still overconfident. For Trajectory Sampling (Fig. 8(c)), increasing the number of models leads to a marginal change in uncertainty and saturates after four models.

Table 11: Performance on proposed metrics at different numbers of keys in memory, with and without data shuffling. The results are averaged over three runs.

| Keys in Memory | Metric | Lotka-Volterra | | FHN | |
|---|---|---|---|---|---|
| | | $\sigma = 0.05$ | | $\sigma = 0.05$ | |
| | | Shuffled Data | Non-Shuffled Data | Shuffled Data | Non-Shuffled Data |
| 10 | MSE | $26.86 \pm 4.80$ | $33.09 \pm 0.009$ | $0.15 \pm 0.03$ | $0.16 \pm 0.01$ |
| | PI-Width | $3.61 \pm 1.06$ | $0.54 \pm 0.001$ | $0.24 \pm 0.03$ | $0.17 \pm 0.005$ |
| | CE | $0.10 \pm 0.04$ | $0.98 \pm 0.007$ | $0.24 \pm 0.19$ | $1.15 \pm 0.05$ |
| 50 | MSE | $9.66 \pm 1.51$ | $32.91 \pm 0.03$ | $0.072 \pm 0.001$ | $0.12 \pm 0.02$ |
| | PI-Width | $2.90 \pm 0.03$ | $0.49 \pm 5.42$ | $0.21 \pm 0.012$ | $0.18 \pm 0.01$ |
| | CE | $0.038 \pm 0.03$ | $0.74 \pm 0.005$ | $0.12 \pm 0.02$ | $0.38 \pm 0.06$ |
| 100 | MSE | $7.76 \pm 0.67$ | $32.63 \pm 0.09$ | $0.076 \pm 0.001$ | $0.10 \pm 0.02$ |
| | PI-Width | $2.99 \pm 0.06$ | $0.51 \pm 0.001$ | $0.21 \pm 0.01$ | $0.16 \pm 0.01$ |
| | CE | $0.026 \pm 0.006$ | $0.72 \pm 0.001$ | $0.111 \pm 0.003$ | $0.46 \pm 0.08$ |
| 500 | MSE | $6.89 \pm 0.30$ | $42.14 \pm 22.03$ | $0.076 \pm 0.002$ | $0.12 \pm 0.03$ |
| | PI-Width | $2.85 \pm 0.08$ | $3.32 \pm 0.09$ | $0.21 \pm 0.006$ | $0.18 \pm 0.009$ |
| | CE | $0.022 \pm 0.003$ | $0.12 \pm 0.06$ | $0.093 \pm 0.03$ | $0.41 \pm 0.09$ |
| 1000 | MSE | $7.08 \pm 0.01$ | $34.80 \pm 16.27$ | $0.077 \pm 0.001$ | $0.092 \pm 0.01$ |
| | PI-Width | $2.91 \pm 0.093$ | $3.36 \pm 0.07$ | $0.21 \pm 0.001$ | $0.27 \pm 0.006$ |
| | CE | $0.021 \pm 0.004$ | $0.083 \pm 0.053$ | $0.092 \pm 0.01$ | $0.12 \pm 0.12$ |
| 2000 | MSE | $6.87 \pm 0.32$ | $60.84 \pm 11.17$ | $0.076 \pm 0.002$ | $0.091 \pm 0.013$ |
| | PI-Width | $2.93 \pm 0.17$ | $4.77 \pm 0.10$ | $0.20 \pm 0.005$ | $0.37 \pm 0.005$ |
| | CE | $0.022 \pm 0.004$ | $0.063 \pm 0.015$ | $0.091 \pm 0.004$ | $0.16 \pm 0.042$ |

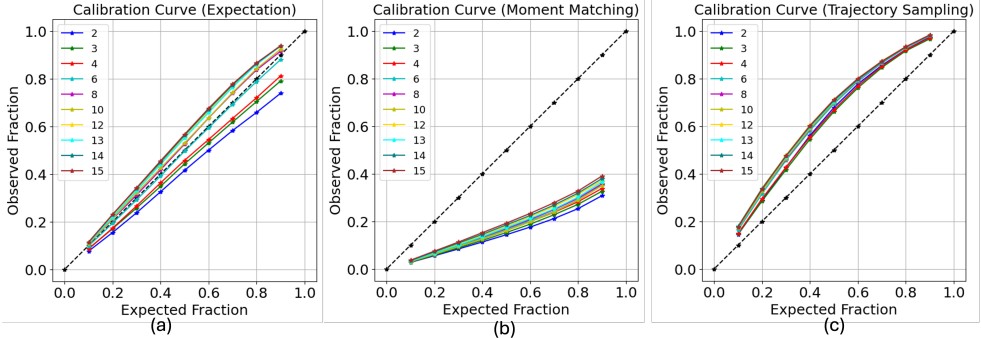

Figure 8: The calibration curves of three propagation approaches for x-output of Lorenz ($\sigma = 0.05$) to delineate the saturation of uncertainty as the number of models in an ensemble goes up from 2 to 15.

### E.9   Sequence length ($S_L$) & Calibration

The section 7 discusses the impact of varying $S_L$ on the model's confidence for the y-output of Lorenz. Here, we show the same for all outputs of the two systems, i.e., Lorenz and Glycolytic Oscillator. The calibrated confidence of both systems with their optimal $S_L$ is shown in the middle plot of Fig. 9a & 9b. To show underconfidence and overconfidence, we pick a small $S_L = 10$ and a large $S_L = 1000$. At small $S_L = 10$, we expect all outputs of both systems to be overconfident as shown in the leftmost plots of Fig. 9a & 9b. At

large $S_L = 1000$, we expect all outputs of both systems to be underconfident as shown in the rightmost plots of Fig. 9a & 9b. It shows that the concept of the increase in uncertainty with the increase in $S_L$ generalizes well across different outputs of different systems.

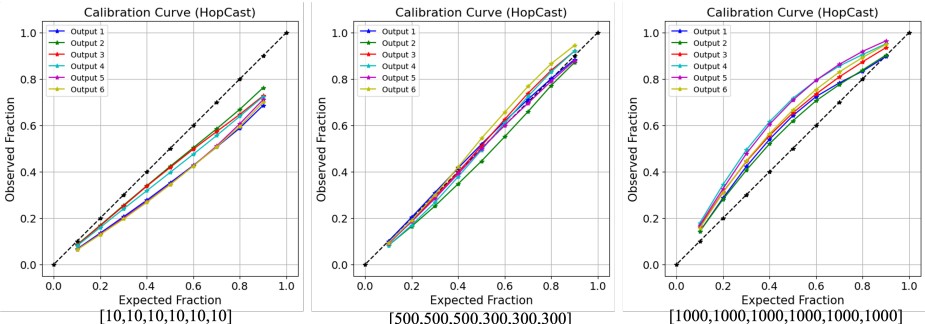

(a) Lorenz ($\sigma = 0.1$). The leftmost plot shows the overconfidence of all six outputs of Lorenz at small $S_L$. The middle plot shows the calibrated confidence for all outputs at optimal $S_L$. The rightmost plot shows the underconfidence of all outputs at large $S_L$.

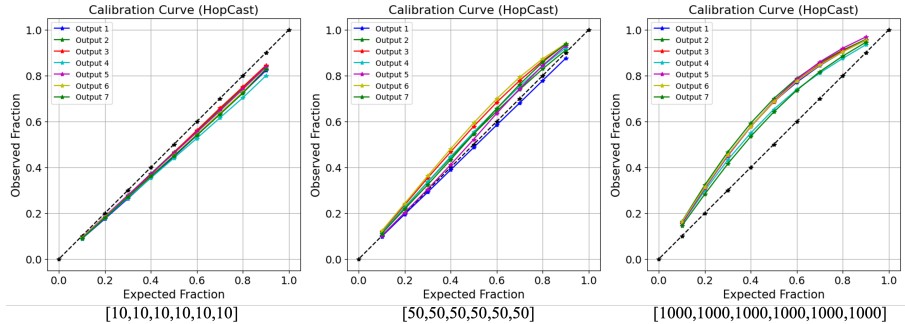

(b) Glycolytic Oscillator ($\sigma = 0.3$). The leftmost plot shows the overconfidence of all seven outputs at small $S_L$. The middle plot shows the calibrated confidence for all outputs at optimal $S_L$. The rightmost plot shows the underconfidence of all outputs at large $S_L$.

Figure 9: Calibration curves of HOPCAST for two dynamical systems to demonstrate the impact of $S_L$ on calibration. At the bottom of each plot, $S_L$ of all outputs are shown in ascending order such that the first value corresponds to the first output, second to the second output, and so on.

# F Implementation Details

This section contains details regarding the implementation of baselines and HOPCAST. All experiments were run on NVIDIA A100-SXM4-80GB. `PyTorch` is used to implement baselines and HOPCAST.

## F.1 Baselines Implementation

We train a population of 15 models for the Probabilistic Ensembles. Regarding architecture, two layers of a fully connected feedforward model with 400 neurons each were used for LV and FHN, and three layers with 400 neurons for Lorenz, Lorenz95, and Glycolytic Oscillator. The train/test datasets were prepared following an 80/20 split. The batch size, learning rate, optimizer, and epochs were kept the same for all experiments, and are 128, 0.001, `Adam` (Kingma & Ba, 2017), and 1000, respectively. The early stopping was used as a criterion to train the Probabilistic Ensembles (Yao et al., 2007). The number of particles $P$ for each model in Probabilistic Ensembles was 1 for Expectation, and 20 for Moment Matching & Trajectory Sampling (Chua et al., 2018a).

### F.2 HopCast Implementation

The Encoder was a fully connected feedforward model with one layer and 100 neurons. The section 7 contains details about tuning an important hyperparameter, i.e., $S_L$, of HOPCAST. Table 3 has $S_L$ for each output of all systems at various noise scaling factors $\sigma$. The learning rate of 0.001 and the optimizer `AdamW` (Loshchilov & Hutter, 2019) were kept the same for all experiments. The batch size and epochs were different for each experiment, and are provided in the form of `yml` files as a supplementary material along with other hyperparameters for each system and noise scaling factor ($\sigma$).

The deterministic Predictor was a fully connected feedforward model of two layers with 400 neurons each for LV and FHN, and three layers with 400 neurons for Lorenz, Lorenz95, and Glycolytic Oscillator.

