# OpenReview forum: "HopCast: Calibration of Autoregressive Dynamics Models"
_TMLR — Accepted by TMLR_

### Review · Reviewer_BLxi · 2025-09-28

**Summary Of Contributions:**

The paper proposes HopCast, a predictor–corrector framework for multi-step forecasting in dynamical systems. A standard autoregressive predictor produces point forecasts, while a learned corrector retrieves context-conditioned historical residuals using an encoder with a Modern Hopfield Network (MHN). The retrieved residuals both (i) correct the point prediction and (ii) yield calibrated prediction intervals (PIs) without explicit uncertainty propagation. A single hyperparameter (“Sequence Length”, SL) controls an effective attention span that trades off sharpness vs. coverage. Experiments on several synthetic ODE systems show improved calibration error with competitive MSE versus ensemble-based baselines and alternative uncertainty propagation schemes. Ablations study memory size, encoder depth, embedding dimension, initial-condition context, and SL.

**Audience:**

Yes

**Audience Explanation:**

The paper is related to dynamical systems and sequential learning, which has a broad audience in TMLR.

**Claims And Evidence:**

Yes

**Claims Explanation:**

- Clear, modular idea: using an associative memory (MHN) to retrieve residuals conditioned on state, initial condition, and time is simple and broadly applicable to autoregressive models.
- Strong focus on calibration: reports calibration error across multiple PI levels, PI width (sharpness), and reliability diagrams; demonstrates a practical knob (SL) to tune the sharpness–coverage tradeoff.
- Useful ablations: thoughtful analysis of architectural/memory choices and the role of initial-condition context; results are interpretable and generally consistent.
- Reasonable path to practice: can be attached to existing predictors without modifying the forecasting backbone.

**Requested Changes:**

*1) Questions for the authors*
- Can the corrector be made multivariate to capture cross-state error covariance? Any preliminary results?
- How sensitive are calibration and MSE to the number of retrieved residual samples and to using top-k or temperature-scaled association weights?
- Could SL (or a temperature on the association distribution) be learned/tuned automatically by minimizing validation calibration error?
- How does HopCast perform under irregular time steps or when tested at Δt different from training?
- What happens under OOD initial conditions or perturbed system parameters (e.g., Lorenz ρ)? Any robustness curves?
- What is the runtime and memory footprint vs. ensemble trajectory sampling, for matched accuracy/calibration?

*2) Suggestions for improvement / additional experiments*
- Implement a joint multivariate corrector (vector-valued residuals) and report CE/MSE and reliability curves vs. the per-dimension approach.
- Add multi-step conformal baselines (rolling residual, naive + recalibration, residual-based with covariates) and a quantile/CRPS-trained sequence model; include lightweight Bayesian last-layer or Laplace/SWAG baselines.
- Automate SL selection (e.g., grid/Bayesian search minimizing validation CE, or learn a softmax temperature as a differentiable knob).
- Analyze sensitivity to residual sampling policy (s, top-k, temperature).
- Test irregular/variable-step scenarios and generalization across Δt.

*3) Minor comments / typos*
- Standardize capitalization (e.g., underconfidence/overconfidence) and hyphenation.
- Ensure consistent notation for vectors/scalars (boldface or arrow).
- Consider moving some algorithmic details and full configs to the supplement to tighten the main text.

---

> ### Author Response · Authors · 2025-10-14
> **comment [1/3]**
>
> **Requested Changes**.
> 1). **Questions for the authors**.
> **Cross-state error covariance**. We thank the reviewer for the suggestion to extend HopCast to a joint multivariate formulation. While this is an interesting direction, such an extension is non-trivial for our retrieval-based framework. HopCast employs Modern Hopfield Networks (MHNs) as dimension-wise correctors, where each MHN learns an associative memory from which errors are sampled based on similarity in terms of context states for a single output dimension of the underlying ODE. The MHN update rule (Eq. 5–8) operates on independent query–key–value sets and performs attention-based retrieval separately for each output. Modeling all state variables jointly would require coupling associative memories for all outputs of an underlying ODE and redefining the MHN energy function to support vector-valued retrieval, which is beyond the architectural capacity of the current MHN formulation. Therefore, we adopted the dimension-wise setup described in Section 4, consistent with prior MHN-based calibration methods (e.g., Auer et al. 2023 [1]). We view multivariate coupling of Hopfield memories as an exciting but orthogonal extension that we plan to investigate in future work.
>
> **$S_L$ tuning/learning**. We thank the reviewer for this valuable suggestion. In our initial experiments, temperature tuning on the association distribution was not as effective as tuning $S_L$ for improving calibration. We observed that $S_L$ provides fine-grained control over calibration by directly controlling the attention span of the MHN and thus the confidence. While $S_L$ can indeed be tuned using standard hyperparameter optimization methods such as Bayesian Optimization or grid search (similar to learning rate or network size), Section 7 shows that the model transitions from overconfidence to underconfidence as $S_L$ increases. We therefore proposed Algorithm 1 as a simple and interpretable procedure that exploits this monotonic relationship between $S_L$ and model confidence to achieve calibrated uncertainty efficiently within 5 to 6 trials.
> In HopCast, learning $S_L$ is non-trivial. $S_L$ determines the number of context states used to construct the association matrix in the MHN. Unlike a temperature parameter that rescales the attention weights, $S_L$ changes the dimensionality of the association matrix ($S_L \times S_L$). This makes gradient-based optimization of $S_L$ non-differentiable and difficult to integrate into end-to-end training.
>
> **Irregular timesteps**. HopCast, in its current formulation, is trained on trajectories sampled at a fixed time interval ($\Delta t$) and relies on explicit timestep identifiers ($t = 1, 2, 3, \ldots$) as part of the context state. Consequently, when the model is evaluated at a different $\Delta t$ or on irregularly sampled trajectories, the learned associations between context states and errors may no longer generalize, leading to high calibration error. This limitation arises because the MHN Corrector treats time as a discrete index rather than a continuous variable. To address this, a discretization-invariant Corrector that treats time as a continuous variable would be required. We have discussed this limitation in Section 8 and plan to explore such formulations in future work.
>
> **OOD Conditions**. The proposed Corrector retrieves errors from associative memory based on similarity in terms of context state that explicitly includes the initial condition and the current state at time t. As a result, substantial OOD shifts, either initial conditions outside the training ranges used to generate trajectories or perturbed system parameters (e.g., Lorenz $\rho$) that induce qualitatively different dynamics, can alter the learned associations (similarities) and degrade both calibration and accuracy. We will mention this limitation in section 8.
>
> **References**.
> [1]. Andreas Auer, Martin Gauch, Daniel Klotz, and Sepp Hochreiter. Conformal prediction for time series with
> modern hopfield networks. Advances in Neural Information Processing Systems, 36:56027–56074, 2023.

---

> ### Author Response · Authors · 2025-10-14
> **comment [2/3]**
>
> **Sensitivity w/r/t $s$ & top-$k$**. We have made two additional ablation studies to show the sensitivity of proposed metrics to (i) varying number of residual samples ($s$) from MHN memory and (ii) varying number of top-$k$ samples from the association distribution. In our preliminary experiments, we did not find temperature scaling as effective as the sequence length ($S_L$) for calibration. Therefore, we did not explore this direction further.
>
> Retrieved Samples ($s$). To generate calibrated prediction intervals, a set of $s$ samples is drawn from $\mathbf{a}_{x}^{t}$ as described in section 4.3 in the paper. Here, we analyze the sensitivity of proposed metrics with varying values of $s$  = {100,500,800,1000} for two settings from Table 2 in the paper. The results everywhere else in the paper are reported with $s=1000$. In Table 1 below, it can be seen that the results are largely unchanged as the number of samples varies from 1000 to 500. However, at $s=100$, we see a drop in performance in terms of MSE and CE, suggesting that too few samples limit the diversity of error samples, negatively impacting the performance.
>
> **Table 1:** Comparison of proposed metrics with varying number of retrieved residual samples ($s$) from MHN memory. The different values of $s$ are {1000,800,500,100}. The results are averaged over three runs.
>
> | **Dynamical System** | |  **Glycolytic Oscillator** |  |  | **Lotka–Volterra** |  |
> |-----------------------|:------------------:|:------------------:|:------------------:|:------------------:|:------------------:|:------------------:|
> |    **Noise level**                  |  | **σ = 0.1**  |   |   |  **σ = 0.1**  |   |
> | **Metrics** | **MSE** | **PI-Width** | **CE** | **MSE** | **PI-Width** | **CE** |
> | **1000** | 0.072 ± 0.003 | 0.25 ± 0.007 | 0.018 ± 0.006 | 6.10 ± 0.20 | 1.88 ± 0.05 | 0.0070 ± 0.002 |
> | **800**  | 0.070 ± 0.005 | 0.24 ± 0.009 | 0.018 ± 0.008 | 6.10 ± 0.20 | 1.89 ± 0.03 | 0.007 ± 0.003 |
> | **500**  | 0.071 ± 0.005 | 0.26 ± 0.003 | 0.015 ± 0.008 | 6.13 ± 0.20 | 1.87 ± 0.04 | 0.006 ± 0.005 |
> | **100**  | 0.096 ± 0.005 | 0.18 ± 0.006 | 0.025 ± 0.008 | 7.90 ± 0.17 | 0.69 ± 0.04 | 0.019 ± 0.004 |
>
> top-$k$. As discussed in section 4.3 in the paper, the $s$ samples are drawn from $\mathbf{a}\_{x}^{t}$ to generate calibrated prediction intervals. Here, we analyze an alternative approach to sampling, i.e., using top-$k$ probabilities from $\mathbf{a}_{x}^{t}$. Table 2 shows the results on proposed metrics with varying values of $k$ = {100,500,800,1000}. The results remain largely unchanged for $k \geq 500$. At $s=100$, the CE shows an increase for both settings, indicating that too few samples lead to decreased diversity in sampled errors.
>
> **Table 2:** Comparison of proposed metrics based on top-*k* probabilities from $\mathbf{a}_x^t$ instead of sampling.
> The results are reported for $k$ = {100, 500, 800, 1000}. The results are averaged over three runs.
>
> | **Dynamical System** |   | **Glycolytic Oscillator** |   |   | **Lotka–Volterra** |   |
> |-----------------------|:------------------:|:------------------:|:------------------:|:------------------:|:------------------:|:------------------:|
> | **Noise Level** | | **σ = 0.1** | | | **σ = 0.1** | |
> | **Metrics** | **MSE** | **PI-Width** | **CE** | **MSE** | **PI-Width** | **CE** |
> | **1000** | 0.67 ± 0.002 | 0.24 ± 0.007 | 0.011 ± 0.005 | 7.31 ± 0.99 | 2.38 ± 0.33 | 0.016 ± 0.003 |
> | **800**  | 0.067 ± 0.002 | 0.24 ± 0.007 | 0.011 ± 0.005 | 7.52 ± 1.18 | 2.41 ± 0.38 | 0.014 ± 0.004 |
> | **500**  | 0.069 ± 0.003 | 0.24 ± 0.009 | 0.017 ± 0.005 | 8.39 ± 2.50 | 2.53 ± 0.64 | 0.017 ± 0.008 |
> | **100**  | 0.07 ± 0.004 | 0.20 ± 0.005 | 0.06 ± 0.03 | 6.36 ± 0.22 | 1.24 ± 0.11 | 0.25 ± 0.036 |
>
> ****

---

> ### Author Response · Authors · 2025-10-14
> **comment [3/3]**
>
> **runtime/memory footprint**.
> **Training time**. HopCast trains one deterministic Predictor and a light Encoder + MHN per output. The Encoder is a fully connected feedforward model with one layer containing 100 neurons. For the Lorenz system, the training time of HopCast is 2 minutes.  On the other hand, the deep ensembles train $M$ probabilistic models where each model is a fully connected feedforward network with 3 layers consisting of 400 neurons each. To train a deep ensemble of 6 models for Lorenz, it took 4 minutes. Early stopping was used to stop training for both cases. The hardware details are mentioned in section E.
> **Inference time.** We report inference time on the entire validation split for the Lorenz system ($\sigma=0.1$), consisting of 200 trajectories of 300 timesteps each. For 7 models with trajectory sampling and 20 particles per model in an ensemble, the deep ensemble took 3.1 seconds to generate calibrated prediction intervals. On the other hand, the HopCast with $K (=2000)$ samples in memory took 3.7 seconds.
> **Memory footprint**. In terms of memory, we need to store $M$ models in memory for deep ensembles. For HopCast, the number of models (Encoder + MHN) stored in memory scales linearly with the dimension of the system of ODEs. An additional memory cost comes from storing $K$ keys in memory, where each $K$ is an embedding of dimension $d$ with corresponding scalar error values. So, the memory of MHN scales as $O(K.d)$. We have $K = 2000$ and $d = 4$ for all cases.
>
> 2). **Additional Experiments**.
>
> **Joint Multivariate Corrector**. Please refer to **Cross-state error covariance** in “Question for authors”.
>
> **Multi-step conformal baselines**. We thank the reviewer for suggesting additional baselines. However, implementing and validating these methods on our problems is not feasible due to the limited time available for the rebuttal period. We would also require open-source code that is compatible with the datasets used in the paper.
>
> **Automate $S_L$ selection**. Please refer to $S\_L$ **tuning/learning** in “Question for authors”.
>
> **Sensitivity w/r/t $s$ & top-$k$**. Please refer to **Sensitivity w/r/t $s$ & top-$k$** in “Question for authors”.
>
> **Irregular step**. Please refer to **Irregular timesteps** in “Question for authors”.
>
> 3). **Minor typos**.
> **Standardize capitalization**. We appreciate the comment and will standardize capitalization and hyphenation (e.g., “under-confidence”/“over-confidence”) in the final version.
>
> **Consistent notation**. Thank you for noting this. We will ensure consistent notation for vectors (boldface) and scalars (regular font) throughout the paper.
>
> **Algorithmic details**. Thanks for the valuable suggestion. We will consider this in the final version.

---

### Review · Reviewer_h5Ri · 2025-10-05

**Summary Of Contributions:**

The paper presents a method to calibrate multi-step learned dynamical systems. The original motivation comes from the fact that typical dynamical systems are trained with one step transitions, leading to multi-step errors; the paper proposes a method that given a trajectory dataset, trains multi-step calibration models that adjust for multi-step uncertainties. The paper improves over alternative methods that train dynamical models given a fixed dataset.

**Audience:**

Yes

**Audience Explanation:**

Since the paper concerns methods to calibrate for uncertainties in training world models, I am sure some TMLR audience will be interested in this work.

**Claims And Evidence:**

Yes

**Claims Explanation:**

The paper seems to present interesting evidence suggesting that the new method  leads to more calibrated dynamical systems compared to baselines. However, I think the solidity of the technical contributions can be made more solid and detailed.

**Requested Changes:**

=== **problem description and dynamical systems 1-2**===

I am slightly confused about the presentation of dynamical 1-2 as the main examples used in the paper. It is not made explicitly clear that the HopCast method is generally applicable to generic dynamical systems, or to systems specific to the form of 1-2. It does seem to me that the method is more general than motivated in Section 3. I'd suggest that the authors present the message in the more clear way, indicating explicitly how general the method is and what algorithmic assumptions and data assumptions are made in the first place. Dynamical system in the form of 1-2 are quite restrictive and I am not sure how interesting they are to general ML audience.

=== **section 4**===

It seems to me that the paper can benefit from a more clear exposition of the content in Section 4, where to me the algorithmic idea and architectural designs (transformer specific) are blended together in a confusing way. Specifically the training procedure in Sec 4.2 does not seem to be directly dependent on the assumption of training a transformer model, instead one can also leverage RNN style state space models. It is not clear to me how much of the algorithmic developments depend on the assumption of using transformers and why is it necessary. Do you think transformer is somehow more advantageous than RNN? On the set of tasks presented in the paper, I am not sure significant difference will be incurred if you repeat the experiments with RNN.

=== **comparison to baselines**===

Though it is definitely interesting to see promising results such as in Fig 5 where true trajectories are within the CI of the prediction, as well as the fact that in Fig 6 the proposed method is more calibrated. Overall I find the paper to lack technical solidity in its comparison against reasonable baselines. After all, calibrated prediction through posterior prediction is well studied in the probabilistic inference literature, and there does not seem to be properly developed baselines compared against in this paper. This puts into question how much gains come from just applying basic posterior inference techniques to the prediction problem itself, compared to HopCast which is designed specifically for this setup (in combination with transformer, a specific architecture).

It is also not clear to me that the method is scalable to higher dimensional more complex setups. More experiments in the context of control-based deep RL environments are very appreciated in this case, rather than low-d ODE simulations.

The paper also lacks ablations in the designs: e.g., why transformer? why not other architectures, etc.

---

> ### Author Response · Authors · 2025-10-14
>
> **Problem description and dynamical systems**.  We thank the reviewer for this comment. The systems in Eqs. (1–2) are used only to illustrate our problem setup and algorithmic details in Section 4 in an easy-to-follow manner. HopCast itself is general and not restricted to that form. Our problem formulation is general and covers any multivariate ordinary differential equation. We will address this in the final version.
> **Regarding exposition in section 4.** We understand the source of confusion, which likely arises from the use of the attention mechanism. Our algorithmic framework is based on the Modern Hopfield Network (MHN), which is a pattern retrieval algorithm. It uses the attention mechanism to retrieve key patterns from its memory, similar to the query pattern. We address this in the first paragraph of Section 4, where we explicitly state our assumption of formulating the correction in our work as a pattern retrieval task. Our methodology does not rely on transformer architecture because we did not formulate the correction in our paper as a sequence modeling task. Hence, RNN is not a suitable alternative to our approach. The motivation to formulate correction as a pattern retrieval task and use MHN is addressed in the third paragraph of the Introduction section.
> **Comparison to baselines**. We appreciate the suggestion to add posterior inference techniques as baselines. We have discussed Bayesian methods in Section 2 of the paper. In the same section, the second paragraph on Ensemble Methods justifies our choice of Deep Ensembles as a baseline. Wilson et al. 2020 [1] compare the performance of deep ensembles and Bayesian posterior inference methods in terms of uncertainty quantification. The paper concludes with the deep ensembles as a scalable and computationally efficient way of quantifying uncertainty compared to posterior inference techniques.
> **More Experiments**. We appreciate the reviewer’s suggestion to include experiments in deep RL settings. We view deep RL as an important application area of our methodology. However, the primary focus of the paper was to validate our proposed methodology on simulated datasets. We plan to explore the integration of HopCast within control-based deep RL environments in future work.
> **Why transformer**. We have addressed this in another answer. Please refer to our answer to the following question: **Regarding exposition in section 4**.
> **References**.
> [1]. Andrew G Wilson and Pavel Izmailov. Bayesian deep learning and a probabilistic perspective of generalization. Advances in neural information processing systems, 33:4697–4708, 2020.

---

### Review · Reviewer_hNjw · 2025-10-07

**Summary Of Contributions:**

This paper proposes an algorithm for calibrating confidence intervals in autoregressive dynamics models with many trajectories. The key to the proposed algorithm is the Modern Hopfield Networks (MHN), which provides samples of the error distribution of these dynamic models. Intensive experiments were conducted to validate the calibration of the proposed algorithm. Various aspects, such as the coverage rate and hyperparameter tuning.  The results seem promising, compared with the basic baselines.

**Additional Comments:**

I have some questions as follows:

- The authors used only $(\bar{x}_1^n, \bar{y}_1^n , \bar{x}_t^n , \bar{y}_t^n , t).$ However, more timestamps can be used. Did you consider this or have any reason for your choice?
- In Fig. 7, the fraction value is limited in the algorithm’s results. Can you provide a finer grid in the $x$-axis, especially near 0 or 1?
- Do you have any idea for the time-series with one trajectory, common in the real world?

**Audience:**

Yes

**Audience Explanation:**

The use of MHN is interesting, and it can provide novel aspects of DNN.

**Broader Impact Concerns:**

This study can affect the advancement of the explainable and accountable AI, which is crucial of the safety and trust AI.

**Claims And Evidence:**

Yes

**Claims Explanation:**

The formulation and simulation setups are solid and sufficiently validated.

**Requested Changes:**

- The role of MHN is a key factor in this paper. Therefore, the motivation, role, and detailed analysis of MHN should be required.
- The prediction of $m$ is too short, which can affect the calibration performance. It requires a more intensive analysis of these aspects.
- The effects of $ \sigma$ and $L$ appear to be non-negligible. Can you provide any suggestions based on the data-driven approaches?

---

> ### Author Response · Authors · 2025-10-14
>
> **Requested Changes**.
> **Role of MHN.** We appreciate the suggestion to address the motivation to use MHN. We addressed this in the third paragraph of the Introduction section. We would be happy to further elaborate on any specific aspects the reviewer finds unclear or missing.
> **Prediction of $m$.** We used the letter m to refer to the Encoder used with MHN. It would be helpful if the reviewer could elaborate further on this comment so that we could address the concern more precisely.
> **Effects of $\sigma$ and $L$**. The sigma denotes the different levels of noise to perturb the system state. These were introduced to see the robustness of the proposed methodology against different levels of observation noise in the system states. The letter L was used as a part of the hyperparameter sequence length ($S_L$), which is a key hyperparameter to get the calibrated prediction intervals. Therefore, it impacts the performance and must be tuned.
> The reviewer referred to the letter L in the comment, but we have answered our question with reference to $S_L$. It would be helpful if the reviewer could revise the comment for clarity.
> **Additional Comments**.
> **More timesteps**. We thank the reviewer for an interesting question. Indeed, other timesteps can be added to the context state. However, our initial experiments showed better performance with the initial condition, the current state, and the timestep ID. Therefore, it was chosen as a context state.
> **Fraction Value**. In Fig. 7, we selected 9 equally spaced prediction intervals from 10% to 90% following Lakshminarayanan et al. 2017 [1] as discussed in section 5.1.  That is why the fraction values range from 0.1 to 0.9 in Fig. 7.
> **One Trajectory**. Yes, our approach will work with such datasets as well. The time series with one trajectory can be split into multiple time series (or trajectories) with a rolling window approach. This will create an appropriate dataset for our proposed methodology.
> **References**.
> 1. Balaji Lakshminarayanan, Alexander Pritzel, and Charles Blundell. Simple and scalable predictive uncertainty estimation using deep ensembles. Advances in neural information processing systems, 30, 2017.

---

> > ### Comment · Reviewer_hNjw · 2025-10-15
> > **Short Response.**
> >
> > 1. Effects of $\sigma$ and $L$.
> > I mean the $L$ in the $\mathcal{S}_L$, and the answer is sufficient to me.
> > 2. More timesteps.
> > Validation from experiments can be right. However, can you provide any regime to make this phenomenon?
> > 3. Prediction of $m$: Sorry for confusing. My main question is about the mean trend of time series, since most content is devoted to the behavior of error terms.
> > 4. Role of MHN: There are no unclear points. However, the introduction of MHN is too brief to fully understand the MHN, including its architecture, learning process, and other aspects.

---

> > > ### Author Response · Authors · 2025-10-17
> > >
> > > 1. **More timesteps.** We appreciate the reviewer’s follow-up question. The observed phenomenon, where adding more timesteps to the context state does not improve calibration or accuracy, arises in regimes where the underlying dynamics are approximately Markovian. In such cases, the current state ($x_t, y_t$) and time index $t$ are sufficient to provide the context state to MHN while adding initial condition provides a small performance boost as shown in Appendix D.2. HopCast works by retrieving the patterns from MHN memory based on similarity in terms of context states, and adding more timesteps to the context state makes the similarity less generalizable on unseen trajectories. This leads to poor calibration performance on validation data. Hence, in Markovian systems, the context state comprising the initial condition, current state, and time identifier yields better performance than longer temporal windows.
> > > 2. **Prediction of $m$**. Thanks for the clarification. We did not focus on the mean trend of time series in the paper because HopCast can correct the Predictor’s output only up to the timesteps seen during training. For example, if the maximum timestep of the trajectory (or time series) in training data is 300, it can only correct the trajectory generated by the Predictor up to 300 timesteps at inference time. We have discussed this limitation in the conclusion section.
> > > 3. **Role of MHN.** We thank the reviewer for this valuable feedback. The role of the MHN is to serve as an associative memory that retrieves stored residual patterns based on similarity in the context state. Since MHN is a well-established architecture, we referred readers to Ramsauer et al. (2021) [1] for full architectural details and focused our discussion on how it is adapted for calibration within HopCast. Specifically, we outlined its role in Section 4 along with Figures 1 & 2. We provided the equations (Eqs. 5–8) describing the retrieval mechanism specifically for our proposed methodology.
> > > **Reference**
> > > [1]. Hubert Ramsauer, Bernhard Schäfl, Johannes Lehner, Philipp Seidl, Michael Widrich, Thomas Adler, Lukas
> > > Gruber, Markus Holzleitner, Milena Pavlović, Geir Kjetil Sandve, et al. Hopfield networks is all you need.arXiv preprint arXiv:2008.02217, 2020.

---

> > > > ### Comment · Reviewer_hNjw · 2025-10-19
> > > > **Reply**
> > > >
> > > > Thanks for your fast reply, and some issues have been resolved. The first issue's answer looks somewhat ad hoc, and the second issue's mean usually cannot be simplified in many cases. Anyway, the response is sufficient for the constructive discussion.

---

### Decision · Action_Editor_7Bjn · 2025-11-18

**Recommendation:** Accept with minor revision

**Additional Comments:**

I agree that this paper is interesting. However, I also agree with Reviewer h5Ri that this paper has some major limitations. Specifically:

1) As Reviewer h5Ri has pointed out, the proposed method is low-dimensional and heavily model-based. Moreover, there is no attempt at combining it with more "modern" RL or control environments to illustrate gains.

2) Overall, this paper seems to lack technical depth. In addition, my understanding is that parts of the paper, especially Section 3 and 4, are not well written.

Some suggestions for final revisions:

1) Please make all the revisions promised in the author responses below.

2) If possible, please discuss how to extend the proposed method to higher dimensional settings.

3) Please improve the writing of Section 3 and 4.

**Audience:**

Yes

**Audience Explanation:**

All three reviewers have agreed that some individuals in TMLR's audience will be interested in knowing the findings of this paper. After reading the paper, I also agree with the reviewers.

**Claims And Evidence:**

Yes

**Claims Explanation:**

All three reviewers have agreed that the claims made in the submission are supported by accurate, convincing, and clear evidence. After reading the paper, I agree with the reviewers.

---

> ### Author Response · Authors · 2025-12-11
> **Revised Manuscript**
>
> We thank the reviewers and the Action Editor for their thoughtful and constructive feedback, which significantly improved the manuscript. Below, we summarize the revisions. All changes are highlighted in red in the revised manuscript.
> - We have addressed the minor typos identified by the reviewer BLxi.
> - We have improved the exposition of sections 3 & 4.
> - The reviewer h5Ri asked for results on control tasks, particularly in model-based RL. Although we consider this a promising future direction, we have added a few preliminary results on the control tasks, demonstrating the performance of HopCast in the context of model-based RL. (Section 8)
>
> We encourage the reviewers and Action Editor to review the revised manuscript and let us know if any additional changes are required. Once the revisions are finalized, we will submit the deanonymized (camera-ready) version.